# Federated Deblurring as a Jigsaw Puzzle: A Privacy-Preserving Consensus Framework

## Abstract

Multi-observation deblurring seeks to recover a clear image from multiple blurred observations, yet most methods assume centralized access to full-scene data, which is unrealistic in practical scenarios where images are captured by distributed, independent devices. We introduce a more realistic federated setting where each client holds a private, partially overlapping view of a larger scene. This creates an "information jigsaw puzzle" that must be solved under strict privacy and regulatory constraints, without sharing raw images, kernels, or intermediate image estimates. We propose `FedDeblur`, a principled federated optimization framework based on consensus that decouples client-side local data fidelity updates and a server-side global image prior update. Clients transmit only carefully designed, desensitized variables, while the server coordinates global consensus. The modularity of our framework enables the server to flexibly incorporate diverse image priors, bridging classic regularizers like total variation with modern deep plug-and-play (PnP) denoisers, all transparently to clients. Crucially, all client-side updates admit efficient closed-form solutions, eliminating the need for inner iterations and making our framework practical for resource-constrained edge devices. Experiments demonstrate that `FedDeblur` seamlessly integrates fragmented information from partial views, effectively solving the jigsaw puzzle and achieving performance close to an idealized, non-private centralized oracle.

## 1 Introduction

Image deblurring is a classic inverse problem in digital image processing, which aims to recover a clear image from a blurred observation. Blur arises from lens imperfections, sensor characteristics, and camera or scene motion, and always coexists with random noise. These image degradations not only degrade visual quality but also harms downstream tasks such as classification, detection, and segmentation (Zhang et al., 2022; Aakanksha & Rajagopalan, 2023). Image deblurring therefore has broad practical value in photography, astronomy, medical imaging, and remote sensing (Donath et al., 2024; Feijoo et al., 2025).

Image blur is commonly modeled as a convolution of a clear image with a blur kernel, or point spread function (PSF). Since recovering the image from the observation is ill-posed, the problem is typically addressed by incorporating a regularizer that reflects prior knowledge about the clear image. Classic regularizers include total variation (TV) (Rudin et al., 1992) and its variants (Bredies et al., 2010), as well as priors based on the $\ell_q$ ($q < 1$) norm (Xu et al., 2011; Zuo et al., 2013), and other non-convex functions (You et al., 2019; Ru et al., 2022). More recently, implicit priors learned through deep learning have also been applied to improve image restoration performance (Venkatakrishnan et al., 2013; Zhang et al., 2021a; Wei et al., 2024).

Beyond designing priors for a single image, an alternative strategy exploits multiple observations of the same scene to jointly recover the latent image. When kernels are weakly correlated, each observation carries complementary information; aggregating them strengthens data-driven constraints and reduces reliance on strong priors. Existing multi-observation methods fall into two groups: model-based iterative methods that design priors and solve an objective iteratively (e.g., Sroubek & Milanfar (2011); Zhang et al. (2014)); and deep learning methods that learn an end-to-end mapping from multiple blurry inputs to one clear image (e.g., Aittala & Durand (2018); Oh & Kim (2022)). Despite good results in specific settings, these approaches struggle with generalizable priors and often incur heavy computation and storage, limiting deployment on edge devices.

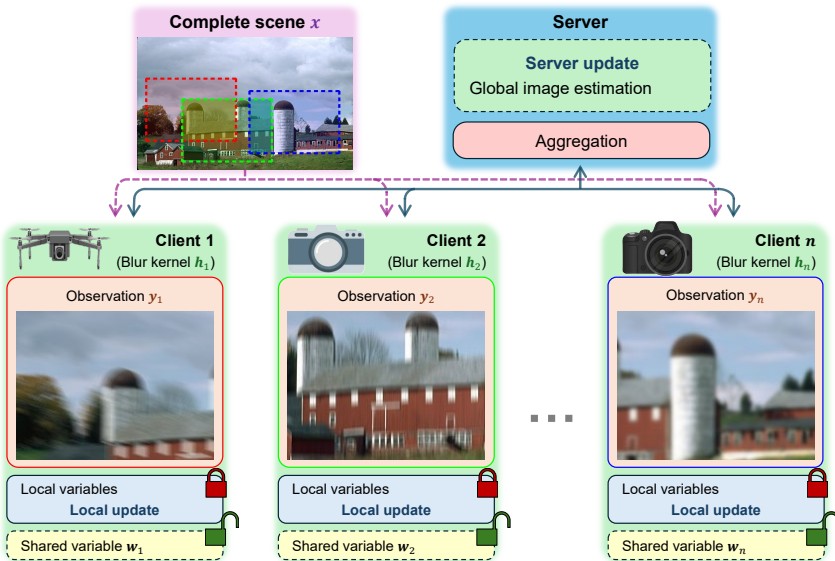

Figure 1: Overview of the proposed federated multi-observation deblurring framework. A central server coordinates multiple clients, each holding a private observation $y_k$ of the scene $x$ and a private kernel $h_k$. Per round, clients update local variables using private data and send only shared variables $w_k$; the server aggregates them to update the global estimate and broadcasts it to guide the next round.

More importantly, most existing works assume centralized processing—all raw data are collected at one node, exposing pixel-level images and kernels. In practice, observations often come from independent devices with distinct hardware and viewpoints, naturally producing different blur kernels. Moreover, a more realistic, challenging case arises when each device captures only a region of the scene, and these regions overlap only partially (Figure 1). Such partially overlapping views form an "information jigsaw puzzle" that breaks the standard assumption of identical full-scene observations.

Furthermore, the degraded observations and their blur kernels can also leak sensitive device or user information. For example, the photo-response non-uniformity (PRNU) is unique to each camera sensor and is widely used as a sensor "fingerprint" (Dirik & Karaküçük, 2014; Banerjee & Ross, 2019), and motion blur can be tied to motion data and hardware (Park & Levoy, 2014; Rong et al., 2024). Therefore, transmitting raw images and kernels may raise significant privacy risks. Moreover, increasingly strict privacy regulations, such as the European Union's General Data Protection Regulation (GDPR), China's Personal Information Protection Law (PIPL), and the EU–US Data Privacy Framework (DPF), legally restrict the arbitrary collection and centralized processing of data.

Federated learning (FL) and federated optimization (McMahan et al., 2017) enable collaborative optimization without sharing raw data, protecting privacy while leveraging diverse sources. This paradigm has already been applied to deep learning (Li et al., 2020a; Banabilah et al., 2022; Huang et al., 2024), low-rank matrix approximation (Nayer & Vaswani, 2022; Abbasi & Vaswani, 2025), and singular value decomposition (SVD) (Chai et al., 2024; Guo et al., 2024). However, to the best of our knowledge, research on multi-observation deblurring has not yet adopted FL principles to achieve high-quality restoration without sharing raw images or kernels. Furthermore, a truly practical solution must also be able to effectively integrate the incomplete information from various participants' partial observations—the "information jigsaw puzzle"—while preserving privacy.

Under the paradigm of federated optimization, we propose an iterative framework for privacy-preserving multi-observation deblurring (Figure 1). Multiple independent *clients* acquire data and communicate only with a central *server*. Clients use private observations to update *local variables* that remain on device; only carefully designed *shared variables* are uploaded for secure aggregation and a global update. With a modular and decoupled design, our framework provides closed-form updates that require no inner iterations, enabling efficient use on edge devices. Our framework strictly adheres to federated principles: (1) no raw observations or kernels are uploaded; (2) clients do not

transmit local clear-image estimates; (3) client updates are lightweight; and (4) in partial observation settings, clients do not reveal their observation regions. Our main contributions are as follows:

- We introduce and formalize the novel problem of federated multi-observation deblurring. In this setting, clients possess private and partially overlapping views of a scene, forming an "information jigsaw puzzle" that must be solved under strict privacy constraints.
- We propose `FedDeblur`, a privacy-preserving consensus framework based on federated optimization. It aggregates fragmented information to reconstruct a high-quality global image without sharing raw images, blur kernels, or intermediate image estimates.
- Our framework is highly modular, allowing seamless adaptation to different observation models (e.g., complete and partial observations) without modifying federated communication protocol. It further enables the server to flexibly integrate diverse image priors—from classic total variation to modern deep plug-and-play models—in a manner that remains transparent to the clients.
- We conduct extensive experiments demonstrating that `FedDeblur` achieves restoration quality close to an idealized centralized oracle, and significantly outperforms both non-collaborative and naive federated baselines, particularly under partial observation settings.

## 2 PROBLEM SETUP

We follow the general setting of federated optimization, adopting a distributed architecture with a central *server* and $n$ *clients* (Ye et al., 2023; Huang et al., 2024). A *client* (indexed by $k$) is an independent device (e.g., camera, mobile device, or other edge systems) that holds its private blur kernel $\boldsymbol{h}_k$ and blurry observation $\boldsymbol{y}_k$. The *server* coordinates computation without accessing client's raw data: it aggregates privacy-preserving summaries from clients, updates a global model, and broadcasts it for the next round.

To facilitate the discussion, we distinguish two types of variables. *Local variables* are computed and stored on device; *shared variables* are intermediate quantities exchanged with the server and designed not to leak sensitive data. Based on privacy considerations and the computational constraints of edge devices, we establish the following federated collaboration principles:

(1) **Data privacy**: Clients never upload raw observations $\boldsymbol{y}_k$ or kernels $\boldsymbol{h}_k$, which may contain sensitive information such as device fingerprints or user behavioral patterns.
(2) **Intermediate privacy**: Clients do not upload local clear-image estimates during iterations, avoiding inference of raw data from image content.
(3) **Computational efficiency**: Local updates are lightweight and simple to implement on resource-constrained edge devices.
(4) **Field-of-view privacy**: In partial observation, clients do not reveal their observed regions as this could reveal sensitive user geolocation data.

We model the multi-observation deblurring problem in two key scenarios.

**Partial observation.** In a realistic federated scenario, the observation regions of different clients often do not fully overlap and may only intersect slightly (as shown in Figure 1). In this case, we model $\boldsymbol{x} \in \mathbb{R}^d$ as a "large" latent image covering all observation areas, and each client's observation $\boldsymbol{y}_k$ corresponds to a local patch of $\boldsymbol{x}$. To formalize this, we introduce a *positional mask* $\boldsymbol{m}_k \in \{0,1\}^d$, known only to client $k$, with ones at its observed pixel locations. This mask embeds the local observation $\boldsymbol{y}_k$ into the full space $\mathbb{R}^d$. The observation model for each client is

$$\boldsymbol{y}_k = \boldsymbol{m}_k \odot (\boldsymbol{h}_k * \boldsymbol{x}) + \boldsymbol{\varepsilon}_k, \quad k = 1, 2, \ldots, n,$$

where $\boldsymbol{h}_k$ and $\boldsymbol{\varepsilon}_k$ are the blur kernel and additive noise for client $k$, respectively, and $\odot$ denotes the element-wise (Hadamard) product. For notational simplicity, we vectorize images, and by an abuse of notation, $*$ denotes 2D convolution with a vectorized input and output. Following the standard modeling approach with data fidelity and regularization terms (Stuart, 2010), we can get

$$\min_{\boldsymbol{x}} \quad \frac{1}{n} \sum_{k=1}^{n} \|\boldsymbol{m}_k \odot (\boldsymbol{h}_k * \boldsymbol{x} - \boldsymbol{y}_k)\|_2^2 + \eta R(\boldsymbol{x}), \tag{1}$$

where $\eta > 0$ balances the fidelity term $F(\boldsymbol{x})$ with the regularization term $R(\boldsymbol{x})$. Note that $\boldsymbol{m}_k$ ensures the loss is computed only on the pixels observed by each client. As private local information, $\boldsymbol{m}_k$ should never be uploaded to the server to satisfy our "field-of-view privacy" principle.

**Complete observation.** This setting follows the basic model used in most existing multi-image deblurring literature. We point out that this scenario, which assumes all clients observe the entire scene, can be treated as a special case of our partial observation model. By setting each positional mask $\boldsymbol{m}_k$ to be a vector of all ones, model (1) reduces to the complete observation model

$$\min_{\boldsymbol{x}} \quad \frac{1}{n} \sum_{k=1}^{n} \|\boldsymbol{h}_k * \boldsymbol{x} - \boldsymbol{y}_k\|_2^2 + \eta R(\boldsymbol{x}). \tag{2}$$

## 3 PROPOSED METHOD

To address challenges posed by federated privacy and partial observation scenarios, we present `FedDeblur`, a federated deblurring framework built on the alternating direction method of multipliers (ADMM) (Boyd et al., 2011). Variable splitting and decoupling align with the "local computation—global aggregation" paradigm and provide a modular basis for more complex scenarios while respecting federated privacy constraints. We first develop the core framework under the complete observation setting, then show that minor modifications extend it to more complex scenarios, including partial observation and transform-domain regularization.

### 3.1 FEDDEBLUR ALGORITHM FRAMEWORK

We base our approach on consensus and federated ADMM (Shi et al., 2014; Tran Dinh et al., 2021) and adapt it to our federated deblurring setting. The objective in (2) is a sum of fidelity terms and a regularizer on a shared variable. We introduce a *local estimate* $\boldsymbol{x}_k$ for each client $k$—a *local variable* updated on device from private data $\{\boldsymbol{y}_k, \boldsymbol{h}_k\}$. A server-side *global estimate* $\boldsymbol{x}$ enforces the consensus constraint $\boldsymbol{x}_k = \boldsymbol{x}$ and aggregates information across clients.

This approach yields the original problem (2) as the following constrained optimization problem:

$$\min_{\boldsymbol{x}, \{\boldsymbol{x}_k\}} \quad \frac{1}{n} \sum_{k=1}^{n} \|\boldsymbol{h}_k * \boldsymbol{x}_k - \boldsymbol{y}_k\|_2^2 + \eta R(\boldsymbol{x}) \quad \text{s.t. } \boldsymbol{x}_k = \boldsymbol{x}, \ k = 1, 2, \ldots, n. \tag{3}$$

This decomposition separates the data-fidelity terms (depending only on each local variable $\boldsymbol{x}_k$) from regularization (depending only on global variable $\boldsymbol{x}$). Its augmented Lagrangian is

$$\mathcal{L}\left(\boldsymbol{x}, \{\boldsymbol{x}_k\}, \{\boldsymbol{\lambda}_k\}\right) = \sum_{k=1}^{n} L_k\left(\boldsymbol{x}, \boldsymbol{x}_k, \boldsymbol{\lambda}_k\right) + \eta R(\boldsymbol{x}),$$

where

$$L_k\left(\boldsymbol{x}, \boldsymbol{x}_k, \boldsymbol{\lambda}_k\right) = \frac{1}{n} \|\boldsymbol{h}_k * \boldsymbol{x}_k - \boldsymbol{y}_k\|_2^2 + \langle \boldsymbol{\lambda}_k, \boldsymbol{x}_k - \boldsymbol{x} \rangle + \frac{\rho}{2} \|\boldsymbol{x}_k - \boldsymbol{x}\|_2^2. \tag{4}$$

Here, $\boldsymbol{\lambda}_k$ is the *local dual variable* associated with the consensus constraint, and $\rho$ is the penalty parameter. The term $L_k$ is the *local objective* for client $k$: it balances data fidelity error based on the client's private data with global consensus via the dual and quadratic terms.

Within the ADMM framework, server and clients alternate minimizing $\mathcal{L}$: the server updates the global variable $\boldsymbol{x}$ and clients in parallel update $\boldsymbol{x}_k$ followed by dual ascent on $\boldsymbol{\lambda}_k$. The overall procedure appears in Algorithm 1 (base option 1 and A), alongside the system schematic in Figure 1.

**Server global update.** The server updates the global estimate by minimizing the global objective $\mathcal{L}$ over $\boldsymbol{x}$. By completing the square, this update is a proximal operator

$$\boldsymbol{x}^{t+1} = \text{Prox}_{\frac{\eta}{n\rho} R} \left( \frac{1}{n} \sum_{k=1}^{n} \left( \boldsymbol{x}_k^t + \frac{\boldsymbol{\lambda}_k^t}{\rho} \right) \right). \tag{5}$$

This is central to our federated design: the server updates $\boldsymbol{x}$ without accessing clients' private data, aggregating only a carefully designed *shared variable*

$$\boldsymbol{w}_k^t = \boldsymbol{x}_k^t + \frac{\boldsymbol{\lambda}_k^t}{\rho}.$$

This shared variable $\boldsymbol{w}_k^t$ combines the local estimate and dual term, obscuring raw information and adhering to the "intermediate privacy" principle.

---

**Algorithm 1** FedDeblur: base , transform-domain regularization and partial observation

---

**Require:** Communication rounds $T$, regularization parameter $\eta$, ADMM parameter $\rho$
 1: **Server**: Initialize $\boldsymbol{x}^0$, $\boldsymbol{z}^0$ and $\boldsymbol{\mu}^0$
 2: **Clients** $k = 1, \ldots, n$: Initialize $\boldsymbol{x}_k^0, \boldsymbol{\lambda}_k^0, \boldsymbol{r}_k^0$ and $\boldsymbol{\beta}_k^0$; compute $\boldsymbol{w}_k^0$ and upload to server
 3: **for** $t = 0, 1, \ldots, T-1$ **do**
 4:    **server do**
 5:       $\boldsymbol{w}^t \leftarrow \frac{1}{n} \sum_{k=1}^n \boldsymbol{w}_k^t$                                                                  ▷ Server aggregate
 6:       *// Option 1: Base: Image-domain regularization*
 7:       $\boldsymbol{x}^{t+1} \leftarrow \mathrm{Prox}_{\eta R/n\rho}(\boldsymbol{w}^t)$
 8:       *// Option 2: Transform-domain regularization*
 9:       $\boldsymbol{z}^{t+1} \leftarrow \mathrm{Prox}_{\eta R/\rho}(\boldsymbol{W}\boldsymbol{x}^t - \boldsymbol{\mu}^t/\rho)$
10:       $\boldsymbol{x}^{t+1} \leftarrow (n\boldsymbol{I} + \boldsymbol{W}^T\boldsymbol{W})^{-1}(n\boldsymbol{w}^t + \boldsymbol{W}^T(\boldsymbol{z}^{t+1} + \boldsymbol{\mu}^t/\rho))$
11:       $\boldsymbol{\mu}^{t+1} \leftarrow \boldsymbol{\mu}^t + \rho(\boldsymbol{z}^{t+1} - \boldsymbol{W}\boldsymbol{x}^{t+1})$
12:       Broadcast $\boldsymbol{x}^{t+1}$ to clients
13:    **for each client $k = 1, \ldots, n$ in parallel do**
14:       *// Option A: Base: Complete observation*
15:       $\boldsymbol{x}_k^{t+1} \leftarrow \mathcal{F}^{-1}\left(\left(\frac{2}{n}\overline{\mathcal{F}(\boldsymbol{h}_k)} \odot \mathcal{F}(\boldsymbol{y}_k) + \mathcal{F}(\rho\boldsymbol{x}^{t+1} - \boldsymbol{\lambda}_k^t)\right)/\left(\frac{2}{n}|\mathcal{F}(\boldsymbol{h}_k)|^2 + \rho\right)\right)$
16:       *// Option B: Partial observation*
17:       $\boldsymbol{r}_k^{t+1} \leftarrow \left(\rho(\boldsymbol{h}_k * \boldsymbol{x}_k^t - \boldsymbol{y}_k) - \boldsymbol{\beta}_k^t\right)/\left(\frac{2}{n}\boldsymbol{m}_k + \rho\right)$
18:       $\boldsymbol{x}_k^{t+1} \leftarrow \mathcal{F}^{-1}\left(\left(\overline{(\mathcal{F}(\boldsymbol{h}_k)} \odot \mathcal{F}(\boldsymbol{r}_k^{t+1} + \boldsymbol{y}_k + \boldsymbol{\beta}_k^t/\rho) + \mathcal{F}(\boldsymbol{x}^{t+1} - \boldsymbol{\lambda}_k^t/\rho)\right)/\left(|\mathcal{F}(\boldsymbol{h}_k)|^2 + 1\right)\right)$
19:       $\boldsymbol{\beta}_k^{t+1} \leftarrow \boldsymbol{\beta}_k^t + \rho(\boldsymbol{r}_k^{t+1} - (\boldsymbol{h}_k * \boldsymbol{x}_k^{t+1} - \boldsymbol{y}_k))$
20:       $\boldsymbol{\lambda}_k^{t+1} \leftarrow \boldsymbol{\lambda}_k^t + \rho(\boldsymbol{x}_k^{t+1} - \boldsymbol{x}^{t+1})$
21:       $\boldsymbol{w}_k^{t+1} \leftarrow \boldsymbol{x}_k^{t+1} + \boldsymbol{\lambda}_k^{t+1}/\rho$ and upload to server                 ▷ Shared variable
22:    **end for**
23: **end for**
24: **return** global estimate $\boldsymbol{x}^T$

---

Note that the server-side update (5) is a proximal step on regularization term $R(\boldsymbol{x})$, the framework naturally accommodates plug-and-play (PnP) priors (Laroche et al., 2023a;b; Wei et al., 2024) by replacing the proximal operator with a pre-trained denoiser $\mathcal{D}_{\eta/n\rho}(\cdot)$, which opens up promising directions for combining our federated framework with state-of-the-art learned priors.

**Client parallel local updates.** Given $\boldsymbol{x}^{t+1}$, each client $k$ minimizes its local objective $L_k$ with respect to $\boldsymbol{x}_k$ in parallel. And this quadratic subproblem admits an efficient closed-form solution in the Fourier domain, which meets our "computational efficiency" requirement for edge devices:

$$\boldsymbol{x}_k^{t+1} = \arg\min_{\boldsymbol{x}_k} L_k(\boldsymbol{x}^{t+1}, \boldsymbol{x}_k, \boldsymbol{\lambda}_k^t)$$
$$= \mathcal{F}^{-1}\left(\frac{\frac{2}{n}\overline{\mathcal{F}(\boldsymbol{h}_k)} \odot \mathcal{F}(\boldsymbol{y}_k) + \mathcal{F}(\rho\boldsymbol{x}^{t+1} - \boldsymbol{\lambda}_k^t)}{\frac{2}{n}|\mathcal{F}(\boldsymbol{h}_k)|^2 + \rho}\right). \tag{6}$$

Each client $k$ then performs dual ascent on $\boldsymbol{\lambda}_k$, forms the shared variable $\boldsymbol{w}_k^{t+1} = \boldsymbol{x}_k^{t+1} + \boldsymbol{\lambda}_k^{t+1}/\rho$, and uploads it.

In each round (see the base options in Algorithm 1), every client solves its private local objective $L_k$ (4) to enforce both data fidelity and consensus, then uploads only the desensitized shared variable $\boldsymbol{w}_k^t$. Thus raw data never leave devices, preserving privacy. The server aggregates $\boldsymbol{w}_k^t$ and performs the proximal prior update (5). Centralizing the prior is appropriate because the scene prior is common across clients and the server's greater computational resources accommodate demanding regularizers (e.g., deep learning-based PnP) without burdening edge devices. This separation of roles—local data fidelity, global prior—produces an efficient design with closed-form local steps and no inner iterations, strictly satisfying all four principles in section 2.

**Theorem 1.** *If $R$ is proper, lower semi-continuous, convex, and bounded below, then for a sufficiently large $\rho$, the sequence of iterates generated by* FedDeblur *algorithm converges to a Karush-Kuhn-Tucker (KKT) point of* (3). *The detailed analysis is provided in appendix E.*

## 3.2 HANDLING PARTIAL OBSERVATIONS

We next address the more challenging problem of partial observations as formulated in (1). The private positional masks $\boldsymbol{m}_k$ in this scenario complicate the data fidelity term and break the Fourier-domain diagonalization of convolution, making the $\boldsymbol{x}_k$-subproblem in (6) nontrivial.

To address this challenge, we exploit the *client-side* modularity of `FedDeblur`. Its client-side modifications appear as option B in Algorithm 1, and detailed derivation is provided in appendix C.1. The core federated consensus mechanism remains unchanged; instead, we introduce a *local* auxiliary variable $\boldsymbol{r}_k = \boldsymbol{h}_k * \boldsymbol{x}_k - \boldsymbol{y}_k$ within each client's local problem. This variable splitting reformulates (1) as

$$\min_{\boldsymbol{x}, \{\boldsymbol{x}_k\}, \{\boldsymbol{r}_k\}} \quad \frac{1}{n} \sum_{k=1}^{n} \|\boldsymbol{m}_k \odot \boldsymbol{r}_k\|_2^2 + \eta R(\boldsymbol{x}) \quad \text{s.t. } \boldsymbol{x}_k = \boldsymbol{x}, \ \boldsymbol{r}_k = \boldsymbol{h}_k * \boldsymbol{x}_k - \boldsymbol{y}_k, \ k = 1, \ldots, n.$$

These additions are purely local, so the complexity stays on the client side; the server update and communication protocol are unchanged and continue to satisfy our federated privacy constraints.

Concretely, the client's local updates in (6) decompose into two simple closed-form steps by introducing an additional local dual variable $\boldsymbol{\beta}_k$:

$$\begin{cases} \boldsymbol{r}_k^{t+1} = \frac{\rho(\boldsymbol{h}_k * \boldsymbol{x}_k^t - \boldsymbol{y}_k) - \boldsymbol{\beta}_k^t}{\frac{2}{n}\boldsymbol{m}_k + \rho}, \\ \boldsymbol{x}_k^{t+1} = \mathcal{F}^{-1}\left( \frac{\overline{\mathcal{F}(\boldsymbol{h}_k)} \odot \mathcal{F}(\boldsymbol{r}_k^{t+1} + \boldsymbol{y}_k + \boldsymbol{\beta}_k^t/\rho) + \mathcal{F}(\boldsymbol{x}^{t+1} - \boldsymbol{\lambda}_k^t/\rho)}{|\mathcal{F}(\boldsymbol{h}_k)|^2 + 1} \right). \end{cases}$$

This extension underscores the modularity of our framework. To handle more complex local observation models (i.e., different data fidelity terms), only the client-side solver needs to be adjusted; the federated architecture and protocol remain unchanged, enabling adaptation to heterogeneous client observation models. Moreover, this variant never uploads the positional mask $\boldsymbol{m}_k$, thus directly enforcing the declared "field-of-view privacy" principle.

## 3.3 INCORPORATING TRANSFORM-DOMAIN REGULARIZATION

Natural images are not sparse in the image domain but often admit compact representations under linear transforms (e.g., gradient, wavelet, Fourier) (Simoncelli & Olshausen, 2001). Consequently, many restoration methods regularize in a transform domain (Zuo et al., 2013; You et al., 2019; Chen & Li, 2025). Let $\boldsymbol{W}$ denote the transform matrix and consider

$$\min_{\boldsymbol{x}} \quad F(\boldsymbol{x}) + \eta R(\boldsymbol{W}\boldsymbol{x}),$$

where $F(\boldsymbol{x})$ is the data fidelity term and $R(\cdot)$ is the transform-domain regularizer. For example, when $\boldsymbol{W}$ is the discrete gradient operator $\boldsymbol{D}$ and $R(\cdot)$ is the $\ell_1$-norm, the regularization is total variation (TV). Although the proximal operator for $R \circ \boldsymbol{W}(\boldsymbol{x})$ (needed in the server update step (5)) is well defined but often lacks a closed form, requiring inner iterations even for simple $R(\cdot)$.

To support such regularizers efficiently, we exploit the *server-side* modularity of our framework. Its server-side modifications are as option 2 in Algorithm 1. We integrate transform regularizers transparently to clients by applying variable splitting at the server. We introduce a *server* auxiliary variable $\boldsymbol{z} = \boldsymbol{W}\boldsymbol{x}$, the clear image in the transform domain. The full derivation of this extension is detailed in appendix C.2.

This change is purely server-side and converts the proximal step (5), which would otherwise require inner iterations, into two simple updates with introducing a new global dual variable $\boldsymbol{\mu}$:

$$\begin{cases} \boldsymbol{z}^{t+1} = \text{Prox}_{\frac{\eta}{\rho}R}\left( \boldsymbol{W}\boldsymbol{x}^t - \boldsymbol{\mu}^t/\rho \right), & (7) \\ \boldsymbol{x}^{t+1} = \left( n\boldsymbol{I} + \boldsymbol{W}^T\boldsymbol{W} \right)^{-1} \left( n\boldsymbol{w}^t + \boldsymbol{W}^T\left( \boldsymbol{z}^{t+1} + \boldsymbol{\mu}^t/\rho \right) \right). & (8) \end{cases}$$

There, (7) decouples the proximal operator of $R \circ \boldsymbol{W}$ into a simpler one involving only $R$, such as soft shrinkage for TV regularization (Daubechies et al., 2004). And when $\boldsymbol{W}$ has a convolutional structure, (8) is efficient in the Fourier domain. For example, for operator $\boldsymbol{D}$ in TV, it becomes

$$\boldsymbol{x}^{t+1} = \mathcal{F}^{-1}\left( \frac{\mathcal{F}\left( n\boldsymbol{w}^t + \boldsymbol{D}^T\left( \boldsymbol{z}^{t+1} + \boldsymbol{\mu}^t/\rho \right) \right)}{n + |\mathcal{F}(\boldsymbol{D})|^2} \right). \quad (9)$$

Crucially, client computation and the communication protocol are unchanged: clients perform the same local updates and upload the same shared variable $w_k$ as before.

Thus, FedDeblur offers a unified, modular framework. In Algorithm 1, blue blocks mark the base framework introduced in section 3.1, green blocks mark the client-side partial observation variant in section 3.2, and red blocks mark the server-side transform regularization variant in section 3.3. Switching among these only replaces the relevant client or server module; the communication pattern and shared variable stay the same. This preserves strict federated principles and keeps computation light (no inner iterations) while still allowing restoration under heterogeneous, partially observed data. Detailed derivations of both extensions appear in appendix C.

## 4 EXPERIMENTS

### 4.1 EXPERIMENTAL SETUP

**Scenarios.** We construct experimental scenarios for both two observation settings:

- **Complete observation.** In this setting, we simulate a scenario where all $n$ clients observe the complete $256 \times 256$ image. For each client $k$, the blur kernel is one of motion, Gaussian, disk, or a random synthesis kernel that simulates camera shake.
- **Partial observation.** In this setting, we partition the complete image into a $3 \times 3$ grid of nine partially overlapping $100 \times 100$ regions, each overlapping adjacent ones by 22 pixels. Each region is the field of view of one of nine clients (see Figure 2a). To model heterogeneous degradations, each client uses an independent blur kernel from one of two families: motion blur with varying angles and lengths (motion) and randomly synthesized camera-shake kernels (synthesis, as shown in Figure 2b, and an example clean scene and its nine observed patches appear in Figure 2c and 2d).

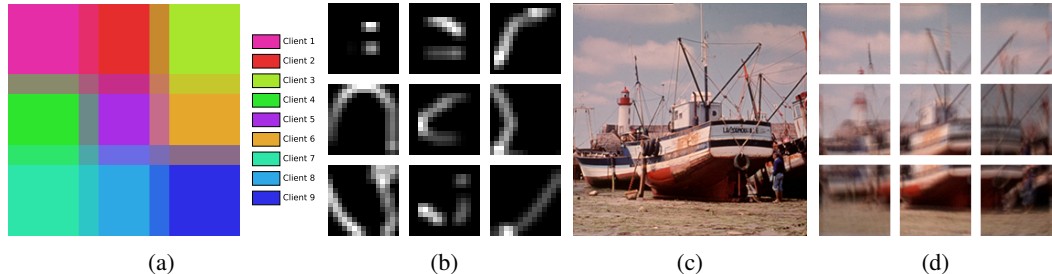

(a)        (b)        (c)        (d)

Figure 2: Visualization of the partial observation setup. We simulate 9 clients, each capturing only a partial, overlapping view of the complete scene. (a) Positional masks: each colored region is one client's field of view. (b) An example set of 9 client-specific blur kernels (synthesis). (c) Example clean scene. (d) The corresponding nine observed patches.

**Baselines.** Our primary goal is to study the effect of the federated consensus architecture itself. Thus, to ensure a fair comparison, we fix the image prior across compared algorithms. We evaluate two priors: classic total variation (TV) regularization and a plug-and-play (PnP) prior instantiated with the DRUNet denoiser (Zhang et al., 2021a). We establish the following baseline methods:

- **Centralized Deblurring (`CenDeblur`)**: A centralized deblurring method that has direct access to all observation data and masks. It solves the global optimization problem (2) or (1) using a classic ADMM algorithm, analogous to traditional methods like Jung et al. (2013); Yang et al. (2013). This method represents an idealized upper bound on performance, assuming privacy and communication constraints are disregarded.
- **Local Deblurring (`LocDeblur`)**: A fully local method where each client independently solves its own deblurring problem using a classic ADMM algorithm (Goldstein & Osher, 2009; Boyd et al., 2011). This method involves no federated collaboration and thus serves as a performance lower bound. The resulting images are averaged only for evaluation purposes.
- **Federated Averaging Deblurring (`FedAvgDeblur`)**: This method is inspired by the federated averaging (`FedAvg`) algorithm (McMahan et al., 2017). Its local iteration framework is similar to `LocDeblur`, but after each local ADMM update, all clients upload their local estimates $x_k$ to the server, which averages them and broadcasts the result as the next round's initialization. Its communication overhead is comparable to that of FedDeblur, but it violates the "intermediate

Table 1: Comparison under TV regularization. Results are presented for complete observation with a varying number of clients ($n = 10, 5, 3$) and partial observation with motion and synthetic blur kernels. For each metric, the best result is highlighted in magenta and the second-best in cyan .

| Setup | Type | FedDeblur | | CenDeblur | | FedAvgDeblur | | LocDeblur | |
|---|---|---|---|---|---|---|---|---|---|
| | | PSNR | SSIM | PSNR | SSIM | PSNR | SSIM | PSNR | SSIM |
| $n = 10$ | Gray | 40.0935 | 0.9780 | 40.1004 | 0.9780 | 39.4586 | 0.9745 | 33.8958 | 0.9363 |
| | Color | 39.8238 | 0.9791 | 39.8283 | 0.9791 | 39.1116 | 0.9755 | 32.7318 | 0.9350 |
| $n = 5$ | Gray | 37.7249 | 0.9661 | 37.7295 | 0.9662 | 37.3823 | 0.9646 | 34.0861 | 0.9406 |
| | Color | 37.4406 | 0.9675 | 37.4436 | 0.9676 | 37.1558 | 0.9655 | 33.2227 | 0.9370 |
| $n = 3$ | Gray | 36.3405 | 0.9571 | 36.3441 | 0.9572 | 36.2079 | 0.9565 | 33.3256 | 0.9365 |
| | Color | 35.9711 | 0.9590 | 35.9714 | 0.9591 | 35.8202 | 0.9574 | 32.3429 | 0.9332 |
| Motion | Gray | 33.7048 | 0.9317 | 33.7041 | 0.9317 | 23.2071 | 0.7298 | 23.2771 | 0.7359 |
| | Color | 32.8611 | 0.9276 | 32.8699 | 0.9277 | 22.5476 | 0.7129 | 22.6383 | 0.7084 |
| Synthesis | Gray | 31.4806 | 0.9121 | 31.6595 | 0.9108 | 21.1035 | 0.6761 | 21.5789 | 0.6656 |
| | Color | 30.5747 | 0.9044 | 30.7922 | 0.8995 | 21.1159 | 0.6606 | 21.2517 | 0.6508 |

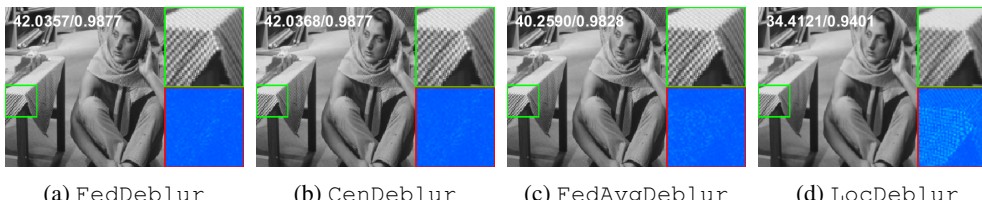

(a) FedDeblur     (b) CenDeblur     (c) FedAvgDeblur     (d) LocDeblur

Figure 3: Visual comparison for the complete observation setting with with $n = 10$ clients, using the PnP prior. Each image is accompanied by an enlarged patch (top right) and its residual map (bottom right). PSNR and SSIM values are displayed on the top left of each image.

   privacy" principle by directly transmitting local image estimates. Furthermore, since each client must independently perform the proximal update, it can be computationally expensive when using deep priors, potentially violating the "computational efficiency" principle.

- **Ours (`FedDeblur`)**: Our proposed federated deblurring method.

Note that for `LocDeblur` and `FedAvgDeblur` in the partial observation setting, the averaging is performed only over the overlapping pixels. As federated algorithms, `FedDeblur` and `FedAvgDeblur` initialize the global estimate $x^0$ randomly, since the server cannot access client data. In contrast, `CenDeblur` initializes $x^0$ with the average of all observed blurry images, i.e., $x^0 = \frac{1}{n} \sum_{k=1}^{n} y_k$. For `LocDeblur`, each client initializes its estimate with its own blurry image, $x_k^0 = y_k$. Detailed information about these baseline algorithms is provided in appendix C.3.

## 4.2 EXPERIMENTAL RESULTS

We report quantitative results for TV and PnP regularization in Table 1 and 2. Each value is the mean metric over multiple grayscale and color test images (see appendix D.1), computed from blurred observations corrupted by additive Gaussian white noise (standard deviation 1). Across all settings, `FedDeblur` achieves performance very close to the idealized centralized method `CenDeblur`, which serves as an upper bound, with only a small gap. This indicates that our framework aggregates information from multiple clients to produce a high-quality global restoration without accessing raw data. Both `FedDeblur` and `CenDeblur` consistently outperform `FedAvgDeblur` and the non-collaborative baseline `LocDeblur`. `LocDeblur`, as the lower bound, yields the worst results, confirming the need to fuse information from multiple observations. The intermediate performance of `FedAvgDeblur` shows that while simple collaboration helps, naive averaging is suboptimal.

In the complete observation setting, the visuals in Figure 3 align with the quantitative results. The collaborative methods—`FedDeblur`, `CenDeblur`, and `FedAvgDeblur`—all recover sharp images with clear edges and fine details, such as the intricate patterns and textures on the tablecloth. In contrast, `LocDeblur` leaves the image blurry, underscoring its inability to exploit complementary information from different clients.

Table 2: Comparison on grayscale images under PnP prior. The experimental settings mirror those in Table 1. The best result is highlighted in magenta and the second-best in cyan .

| Setup | FedDeblur | | CenDeblur | | FedAvgDeblur | | LocDeblur | |
|---|---|---|---|---|---|---|---|---|
| | PSNR | SSIM | PSNR | SSIM | PSNR | SSIM | PSNR | SSIM |
| $n = 10$ | 42.1380 | 0.9850 | 42.1415 | 0.9850 | 40.9416 | 0.9813 | 35.0000 | 0.9503 |
| $n = 5$ | 39.7744 | 0.9737 | 39.7207 | 0.9759 | 37.9846 | 0.9669 | 35.6389 | 0.9543 |
| $n = 3$ | 38.6532 | 0.9714 | 38.7273 | 0.9711 | 37.3461 | 0.9618 | 34.5894 | 0.9493 |
| motion | 34.7655 | 0.9416 | 35.3347 | 0.9458 | 22.7676 | 0.7369 | 22.5329 | 0.6996 |
| synthesis | 31.5206 | 0.9124 | 32.3711 | 0.9164 | 21.2241 | 0.6491 | 20.8160 | 0.5974 |

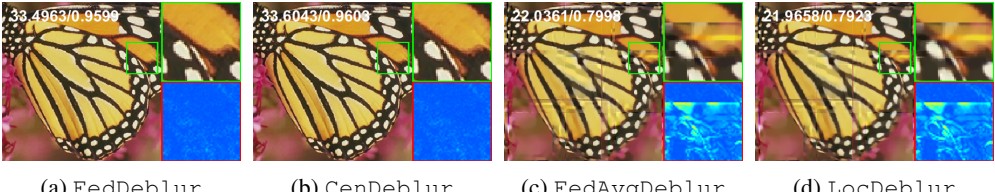

(a) FedDeblur  (b) CenDeblur  (c) FedAvgDeblur  (d) LocDeblur

Figure 4: Visual comparison for the partial observation setting with motion blur, using TV regularization. Each image is accompanied by an enlarged patch (top right) and its residual map (bottom right). PSNR and SSIM values are displayed on the top left of each image.

The advantages of our framework are most pronounced in the challenging partial observation setting. In this case, as shown in Table 1 and 2, the performance of FedAvgDeblur and LocDeblur degrades sharply. Figure 4 further explains this failure: both methods create strong seam artifacts in overlapping regions. Moreover, the corresponding residual maps show structured, block-wise errors, indicating that naive averaging or independent processing cannot reconcile inconsistent fields of view and may even introduce conflicting updates. By contrast, FedDeblur and CenDeblur produce seamless, natural-looking images. This demonstrates that our method effectively integrates information from different spatial regions into a unified global estimate, solving the "information jigsaw puzzle" under strict federated privacy constraints.

Finally, our results highlight the framework's robustness and generality. From Table 1 and 2, increasing the number of clients $n$ from 3 to 10 leads to a consistent improvement for all collaborative methods (FedDeblur, CenDeblur, FedAvgDeblur), since more observations provide complementary constraints that suppress artifacts. Furthermore, although the PnP prior yields better absolute metrics than TV, the performance ranking and relative gaps among the four methods remain stable across priors. Therefore, the performance differences arise from the federated optimization designs rather than the choice of regularizer. Consequently, FedDeblur provides a flexible framework that combines cleanly with diverse image priors, from classic models to learned denoisers.

For a more comprehensive evaluation, we report additional experiments in appendix D. We analyze scalability with respect to the number of clients (appendix D.2), performance under partial client participation (appendix D.3), and robustness to varying levels of observation noise (appendix D.4). We also provide a detailed numerical analysis of convergence behavior (appendix D.5) and sensitivity to hyperparameters (appendix D.8).

## 5 CONCLUSION

We addressed multi-observation deblurring in a privacy-constrained federated setting where clients each observe only partial, overlapping regions of a scene. We introduced FedDeblur, a federated consensus framework that solves this "information jigsaw puzzle" by aggregating fragmented data without compromising privacy while strictly following federated principles. With a modular, decoupled design separating local data fidelity and the global image prior, FedDeblur handles diverse partial observation regimes and supports a wide range of priors, from classical regularizers to deep learning PnP models. Experiments show that the method achieves high-quality restoration close to an idealized non-private centralized solution. Future work will extend the framework to asynchronous client participation and richer forms of data heterogeneity.

REPRODUCIBILITY STATEMENT

We are committed to ensuring the reproducibility of our research. The complete source code for our proposed `FedDeblur`, all baseline methods, and the scripts to replicate our experiments are provided in `https://anonymous.4open.science/r/FedDeblur-53D7`. All experiments use fixed random seeds to ensure deterministic results and were conducted on a workstation with NVIDIA RTX 3090 GPU. The complete mathematical derivations and all proofs are stated in the appendix with explanations and underlying assumptions.

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

## A    RELATED WORK

### A.1    MULTI-IMAGE DEBLURRING

Existing research in multi-image deblurring underpins our problem formulation but has largely assumed centralized access to data. These methods can be broadly divided into model-based iterative optimization and end-to-end deep learning approaches.

Within the framework of model-based iterative optimization, Sroubek & Milanfar (2011) first proposed a robust multi-image deblurring algorithm based on fast alternating minimization. Faramarzi et al. (2013) integrated multi-image super-resolution and deblurring into a unified framework. Zhang et al. (2013; 2014) introduced coupled adaptive sparse priors, which automatically assign weights to each observation based on its intrinsic quality. Zhou et al. (2021) applied multi-frame deconvolution with a hyper-Laplacian prior to rotated rectangular aperture imaging. In contrast to these optimization-based methods, Delbracio & Sapiro (2015) proposed a method that directly performs weighted information fusion in the Fourier domain. However, these iterative methods are computationally expensive and can be limited by handcrafted priors, which may adapt poorly to complex scenes. Our work retains the modeling strengths of these methods but targets deployment on edge devices: all client updates are closed-form, require no inner iterations, and the design remains compatible with diverse priors.

Deep learning methods learn an end-to-end mapping from multiple blurry inputs to a single clear image. Wieschollek et al. (2017) used a recurrent neural network (RNN) to process image bursts by exploiting temporal information. Aittala & Durand (2018) proposed a permutation-invariant convolutional neural network (CNN) architecture capable of processing an arbitrary number of input images in a sequence-agnostic manner. Peña et al. (2019) employed a ranking network to pre-select the sharpest frames from a burst before fusion to improve deblurring performance. Furthermore, Zhou et al. (2019) designed DAVANet for information aggregation in stereo cameras, while Oh & Kim (2022) introduced DeMFI for video data, which jointly performs multi-frame video deblurring and interpolation within a unified framework. These approaches, however, rely on large paired datasets and often have heavy models, making deployment on heterogeneous edge devices difficult. By contrast, our iterative framework does not require large-scale training data and, when available, can seamlessly integrate learned PnP priors.

### A.2    FEDERATED OPTIMIZATION

Our work builds on principles of federated optimization to address privacy. The federated averaging (FedAvg) algorithm of McMahan et al. (2017) is the foundational work in this field. It significantly improves communication efficiency by allowing clients to perform multiple steps of local stochastic gradient descent (SGD) in each communication round. However, in typical federated learning scenarios, data on client devices is typically non-independent and identically distributed (non-IID), leading to what is known as "data heterogeneity" and the "client drift" problem (Ye et al., 2023).

To address this core challenge, researchers have proposed a range of federated optimization algorithms. One line of research modifies the local objective function to constrain local updates and prevent them from deviating too far from the global model. For example, Li et al. (2020b) proposed FedProx, which adds a proximal term to the local loss function to limit the magnitude of local updates. The FedDyn algorithm by Durmus et al. (2021) employs a dynamic regularizer, ensuring that the minimizers of the local objectives asymptotically align with the stationary points of the global loss. In contrast to these methods, SCAFFOLD, proposed by Karimireddy et al. (2020), focuses on directly correcting the direction of the local update gradient.

Another major class of methods leverages operator splitting techniques to provide a more structured mathematical framework for federated optimization problems. Our proposed FedDeblur framework is built upon this line of work, leveraging the consensus form of ADMM, which is a natural fit for the "local computation-global aggregation" paradigm of federated optimization. Zhang et al. (2021b) first proposed the FedPD algorithm under this framework, followed by several FedADMM algorithms from Gong et al. (2022) and Zhou & Li (2023). Additionally, Tran Dinh et al. (2021) proposed the FedDR algorithm based on Douglas-Rachford splitting, which is equivalent to ADMM under certain conditions (Wang et al., 2022). Operator-splitting frameworks offer strong decoupling

and handle non-smooth terms well, making them well-suited to our modular design that separates local data fitting from global regularization.

While initially designed for training deep learning models in federated learning, recent research has successfully extended the federated paradigm to a range of classic iterative optimization algorithms, offering distributed solutions that balance privacy and computational efficiency. Grammenos et al. (2020) designed a differentially private federated PCA algorithm for asynchronous and memory-constrained environments. In the field of low-rank matrix approximation, a strategy of decomposing the target matrix into local and shared factors to decouple local and global updates has been successfully applied to problems like low-rank matrix recovery (Nayer & Vaswani, 2022) and matrix completion (Abbasi & Vaswani, 2025). For the singular value decomposition (SVD) problem, Guo et al. (2024) extended the classic power method to the federated setting. Differing from this iterative approach, Chai et al. (2022; 2024) focused on achieving an exact federated SVD. Furthermore, for high-dimensional data scenarios such as genomics, Hartebrodt et al. (2024) designed a federated SVD algorithm with communication costs independent of the sample size. For non-negative matrix factorization (NMF), Dalleiger & Gionis (2025) proposed a solution based on "creating coherence". Our work develops a federated framework for an image inverse problem—multi—observation deblurring—highlighting the breadth of applications for federated optimization.

# B    NOTATIONS

We summarize the notation used in this paper. Plain lowercase letters denote scalars; bold lowercase letters denote vectors; and capital letters denote matrices and linear operators. Images and related variables are treated as vectors in the $d$-dimensional real Euclidean space $\mathbb{R}^d$ with inner product $\langle \cdot, \cdot \rangle$. The $\ell_2$-norm is denoted by $\|\cdot\|_2$. Two-dimensional convolution is denoted by $*$, and the 2D discrete Fourier transform and its inverse are $\mathcal{F}(\cdot)$ and $\mathcal{F}^{-1}(\cdot)$, respectively. For these 2D operators, although inputs and outputs are written as vectors, we apply the operations to their two-dimensional reshaped forms and then vectorize the results. Element-wise operations include the Hadamard product ($\odot$), division, exponentiation (for any vector $\boldsymbol{v}$ and scalar $p$, $\boldsymbol{v}^p$ denotes element-wise power), complex conjugation (denoted by an overline, e.g., $\overline{\boldsymbol{z}}$), and modulus for complex vectors (for $\boldsymbol{z}$, $|\boldsymbol{z}|$ and $|\boldsymbol{z}|^2$ denote the element-wise modulus and squared modulus). The transpose of a linear operator $\boldsymbol{A}$ is $\boldsymbol{A}^T$, and $I$ denotes the identity matrix. A superscript $t$ (e.g., $\boldsymbol{x}^t$) indicates the iteration index in our algorithms.

# C    MORE ALGORITHMIC DETAILS

This section provides further details on the extensions of the FedDeblur framework, including partial observations and transform-domain regularization. We begin by deriving these extensions and presenting an integrated, unified algorithm. Subsequently, we will elaborate on the implementation details of the baseline algorithms used for comparison in the experiments.

## C.1    FEDDEBLUR FOR PARTIAL OBSERVATION

For the partial observation scenario, the objective function is given by

$$\min_{\boldsymbol{x}} \quad \frac{1}{n} \sum_{k=1}^n \|\boldsymbol{m}_k \odot (\boldsymbol{h}_k * \boldsymbol{x} - \boldsymbol{y}_k)\|_2^2 + \eta R(\boldsymbol{x}).$$

As in the base framework, we introduce local variables $\boldsymbol{x}_k$ and enforce the consensus constraint $\boldsymbol{x}_k = \boldsymbol{x}$. The presence of the positional mask $\boldsymbol{m}_k$ complicates the client subproblem, as it prevents a direct solution in the Fourier domain. To address this, we introduce an additional local auxiliary variable for each client, $\boldsymbol{r}_k = \boldsymbol{h}_k * \boldsymbol{x}_k - \boldsymbol{y}_k$. The problem can then be equivalently reformulated as

$$\min_{\boldsymbol{x}, \{\boldsymbol{x}_k\}, \{\boldsymbol{r}_k\}} \quad \frac{1}{n} \sum_{k=1}^n \|\boldsymbol{m}_k \odot \boldsymbol{r}_k\|_2^2 + \eta R(\boldsymbol{x})$$
$$\text{s.t.} \quad \boldsymbol{x}_k = \boldsymbol{x}, \quad \boldsymbol{r}_k = \boldsymbol{h}_k * \boldsymbol{x}_k - \boldsymbol{y}_k, \quad k = 1, \dots, n.$$

The augmented Lagrangian now includes terms for two sets of constraints:

$$\mathcal{L}\left(\boldsymbol{x}, \{\boldsymbol{x}_k\}, \{\boldsymbol{r}_k\}, \{\boldsymbol{\lambda}_k\}, \{\boldsymbol{\beta}_k\}\right) = \sum_{k=1}^{n} L_k\left(\boldsymbol{x}, \boldsymbol{x}_k, \boldsymbol{r}_k, \boldsymbol{\lambda}_k, \boldsymbol{\beta}_k\right) + \eta R(\boldsymbol{x}),$$

$$\text{where} \quad L_k\left(\boldsymbol{x}, \boldsymbol{x}_k, \boldsymbol{r}_k, \boldsymbol{\lambda}_k, \boldsymbol{\beta}_k\right) = \frac{1}{n}\left\|\boldsymbol{m}_k \odot \boldsymbol{r}_k\right\|_2^2$$
$$+ \langle\boldsymbol{\lambda}_k, \boldsymbol{x}_k - \boldsymbol{x}\rangle + \frac{\rho}{2}\left\|\boldsymbol{x}_k - \boldsymbol{x}\right\|_2^2$$
$$+ \langle\boldsymbol{\beta}_k, \boldsymbol{r}_k - (\boldsymbol{h}_k * \boldsymbol{x}_k - \boldsymbol{y}_k)\rangle + \frac{\rho}{2}\left\|\boldsymbol{r}_k - (\boldsymbol{h}_k * \boldsymbol{x}_k - \boldsymbol{y}_k)\right\|_2^2. \tag{10}$$

Here, $\boldsymbol{\beta}_k$ is the local dual variable associated with the new constraint. The server update for $\boldsymbol{x}$ remains unchanged. The client's local update now involves sequentially solving for $\boldsymbol{r}_k$ and $\boldsymbol{x}_k$ by minimizing the local objective $L_k$.

**Client's local update.** Each client $k$ solves for its local variables by minimizing its local objective $L_k$:

$$\boldsymbol{r}_k^{t+1} = \arg\min_{\boldsymbol{r}_k} L_k\left(\boldsymbol{x}^{t+1}, \boldsymbol{x}_k^t, \boldsymbol{r}_k, \boldsymbol{\lambda}_k^t, \boldsymbol{\beta}_k^t\right),$$
$$\boldsymbol{x}_k^{t+1} = \arg\min_{\boldsymbol{x}_k} L_k\left(\boldsymbol{x}^{t+1}, \boldsymbol{x}_k, \boldsymbol{r}_k^{t+1}, \boldsymbol{\lambda}_k^t, \boldsymbol{\beta}_k^t\right).$$

Both subproblems are quadratic optimization problems with closed-form solutions. Specifically,

$$\boldsymbol{r}_k^{t+1} = \frac{\rho\left(\boldsymbol{h}_k * \boldsymbol{x}_k^t - \boldsymbol{y}_k\right) - \boldsymbol{\beta}_k^t}{\frac{2}{n}\boldsymbol{m}_k^2 + \rho}.$$

Note that since $\boldsymbol{m}_k \in \{0, 1\}^d$, we have $\boldsymbol{m}_k^2 = \boldsymbol{m}_k$. The squaring and division operations here are performed element-wise. Notably, the $\boldsymbol{x}_k$-subproblem is a standard deconvolution problem that can be solved efficiently in the Fourier domain:

$$\boldsymbol{x}_k^{t+1} = \mathcal{F}^{-1}\left(\frac{\overline{\mathcal{F}\left(\boldsymbol{h}_k\right)} \odot \mathcal{F}\left(\boldsymbol{r}_k^{t+1} + \boldsymbol{y}_k + \boldsymbol{\beta}_k^t/\rho\right) + \mathcal{F}\left(\boldsymbol{x}^{t+1} - \boldsymbol{\lambda}_k^t/\rho\right)}{\left|\mathcal{F}\left(\boldsymbol{h}_k\right)\right|^2 + 1}\right).$$

Finally, the dual ascent updates for the local dual variables are standard:

$$\boldsymbol{\lambda}_k^{t+1} = \boldsymbol{\lambda}_k^t + \rho(\boldsymbol{x}_k^{t+1} - \boldsymbol{x}^{t+1}),$$
$$\boldsymbol{\beta}_k^{t+1} = \boldsymbol{\beta}_k^t + \rho(\boldsymbol{r}_k^{t+1} - (\boldsymbol{h}_k * \boldsymbol{x}_k^{t+1} - \boldsymbol{y}_k)).$$

## C.2 FEDDEBLUR WITH TRANSFORM-DOMAIN REGULARIZATION

To handle transform-domain regularization, the objective function takes the form of

$$\min_{\boldsymbol{x}} \quad F(\boldsymbol{x}) + \eta R(\boldsymbol{W}\boldsymbol{x}),$$

where $F(\boldsymbol{x})$ is the task-specific data fidelity term, which can be either $\frac{1}{n}\sum_{k=1}^{n}\|\boldsymbol{h}_k * \boldsymbol{x} - \boldsymbol{y}_k\|_2^2$ or $\frac{1}{n}\sum_{k=1}^{n}\|\boldsymbol{m}_k \odot (\boldsymbol{h}_k * \boldsymbol{x} - \boldsymbol{y}_k)\|_2^2$, and $\boldsymbol{W}$ is a linear transform operator. Directly applying the proximal operator of $R(\boldsymbol{W}\boldsymbol{x})$ on the server can be computationally expensive. Therefore, we introduce a global auxiliary variable $\boldsymbol{z} = \boldsymbol{W}\boldsymbol{x}$ at the server. The federated consensus problem becomes

$$\min_{\boldsymbol{x}, \boldsymbol{z}, \{\boldsymbol{x}_k\}} \quad F\left(\boldsymbol{x}_k\right) + \eta R(\boldsymbol{z})$$
$$\text{s.t.} \quad \boldsymbol{x}_k = \boldsymbol{x}, \quad \boldsymbol{z} = \boldsymbol{W}\boldsymbol{x}, \quad k = 1, \ldots, n.$$

This modification only affects the server-side components and has no impact on the local variables or the local objective. The augmented Lagrangian is

$$\mathcal{L}(\boldsymbol{x}, \boldsymbol{z}, \boldsymbol{\mu}, \{\boldsymbol{x}_k\}, \ldots) = \sum_{k=1}^{n} L_k\left(\boldsymbol{x}, \boldsymbol{x}_k, \ldots\right) + \eta R(\boldsymbol{z}) + \langle\boldsymbol{\mu}, \boldsymbol{z} - \boldsymbol{W}\boldsymbol{x}\rangle + \frac{\rho}{2}\|\boldsymbol{z} - \boldsymbol{W}\boldsymbol{x}\|_2^2.$$

---

**Algorithm 2** `CenDeblur-TV` for complete observation and partial observation

---

**Require:** Max iterations $T$, regularization parameter $\eta$, ADMM parameter $\rho$

1: Initialize $\boldsymbol{x}^0$, $\boldsymbol{z}^0$, $\boldsymbol{\mu}^0$ and $\boldsymbol{\beta}_k^0$ for partial observation
2: **for** $t = 0, 1, \ldots, T-1$ **do**
3:     $\boldsymbol{z}^{t+1} \leftarrow \mathrm{Shrink}_{\eta/\rho}\left(\boldsymbol{D}\boldsymbol{x}^t - \boldsymbol{\mu}^t/\rho\right)$             ▷ Soft shrinkage
4:     *// Complete observation*
5:     $\boldsymbol{x}^{t+1} \leftarrow \mathcal{F}^{-1}\left( \dfrac{\frac{2}{n}\sum_{k=1}^n \overline{\mathcal{F}(\boldsymbol{h}_k)}\mathcal{F}(\boldsymbol{y}_k) + \mathcal{F}\left(\rho\boldsymbol{D}^T(\boldsymbol{z}^{t+1}+\boldsymbol{\mu}^t/\rho)\right)}{\frac{2}{n}\sum_{k=1}^n |\mathcal{F}(\boldsymbol{h}_k)|^2 + \rho|\mathcal{F}(\boldsymbol{D})|^2} \right)$
6:     *// Partial observation*
7:     $\boldsymbol{x}^{t+1} \leftarrow \mathcal{F}^{-1}\left( \dfrac{\sum_{k=1}^n \overline{\mathcal{F}(\boldsymbol{h}_k)}\mathcal{F}(\boldsymbol{r}_k^t+\boldsymbol{y}_k+\boldsymbol{\beta}_k^t/\rho) + \mathcal{F}\left(\boldsymbol{D}^T(\boldsymbol{z}^{t+1}+\boldsymbol{\mu}^t/\rho)\right)}{\sum_{k=1}^n |\mathcal{F}(\boldsymbol{h}_k)|^2 + |\mathcal{F}(\boldsymbol{D})|^2} \right)$
8:     **for** $k = 1, \ldots, n$ **do**
9:        $\boldsymbol{r}_k^{t+1} \leftarrow \dfrac{\rho(\boldsymbol{h}_k * \boldsymbol{x}^{t+1} - \boldsymbol{y}_k) - \boldsymbol{\beta}_k^t}{\frac{2}{n}\boldsymbol{m}_k + \rho}$
10:        $\boldsymbol{\beta}_k^{t+1} \leftarrow \boldsymbol{\beta}_k^t + \rho\left(\boldsymbol{r}_k^{t+1} - (\boldsymbol{h}_k * \boldsymbol{x}^{t+1} - \boldsymbol{y}_k)\right)$
11:     **end for**
12:     $\boldsymbol{\mu}^{t+1} \leftarrow \boldsymbol{\mu}^t + \rho\left(\boldsymbol{z}^{t+1} - \boldsymbol{D}\boldsymbol{x}^{t+1}\right)$
13: **end for**
14: **return** $\boldsymbol{x}^T$

---

Here, $L_k$ is the task-specific local objective (related to the fidelity term) as defined in (4) or (10), and $\boldsymbol{\mu}$ is a global dual variable. Client updates for local variables remain consistent with the image-domain regularization cases described in sections 3.1 and 3.2. The server update for the global primal variables is now split into separate updates for $\boldsymbol{z}$ and $\boldsymbol{x}$.

**Server's global update.** The server performs the same global aggregation as before:

$$\boldsymbol{w}^t = \frac{1}{n}\sum_{k=1}^n \boldsymbol{w}_k^t.$$

The global variables are updated as follows:

$$\boldsymbol{z}^{t+1} = \arg\min_{\boldsymbol{z}} \mathcal{L}\left(\boldsymbol{x}^t, \boldsymbol{z}, \boldsymbol{\mu}^t, \{\boldsymbol{x}_k^t\}, \ldots\right),$$
$$\boldsymbol{x}^{t+1} = \arg\min_{\boldsymbol{x}} \mathcal{L}\left(\boldsymbol{x}, \boldsymbol{z}^{t+1}, \boldsymbol{\mu}^t, \{\boldsymbol{x}_k^t\}, \ldots\right),$$
$$\boldsymbol{\mu}^{t+1} = \boldsymbol{\mu}^t + \rho(\boldsymbol{z}^{t+1} - \boldsymbol{W}\boldsymbol{x}^{t+1}).$$

After some algebraic manipulation, the $\boldsymbol{z}$-subproblem decouples the proximal update, replacing the operator for $R(\boldsymbol{W}\boldsymbol{x})$ with a simpler one for $R(\boldsymbol{z})$:

$$\boldsymbol{z}^{t+1} = \mathrm{Prox}_{\frac{\eta}{\rho}R}\left(\boldsymbol{W}\boldsymbol{x}^t - \boldsymbol{\mu}^t/\rho\right).$$

The $\boldsymbol{x}$-subproblem is a quadratic optimization problem with a closed-form solution

$$\boldsymbol{x}^{t+1} = \left(n\boldsymbol{I} + \boldsymbol{W}^T\boldsymbol{W}\right)^{-1}\left(n\boldsymbol{w}^t + \boldsymbol{W}^T\left(\boldsymbol{z}^{t+1} + \boldsymbol{\mu}^t/\rho\right)\right).$$

If $\boldsymbol{W}$ has a convolutional structure (e.g., the gradient operator), this inverse can be computed efficiently in the Fourier domain, as shown in (9).

## C.3   BASELINE ALGORITHMS

This section provides detailed information on the baseline algorithms used in the experiments (section 4). For each baseline method, we implement two variants: one using total variation (TV) regularization, and another using a plug-and-play (PnP) prior with a pre-trained DRUNet denoiser (Zhang et al., 2021a).

**`CenDeblur`.** This method represents an idealized centralized scenario and serves as an upper-bound benchmark for performance. In this setting, the central server has direct access to all clients'

---

**Algorithm 3** `CenDeblur-PnP` for complete observation and partial observation

---

**Require:** Max iterations $T$, regularization parameter $\eta$, ADMM parameter $\rho$

1: Initialize $\boldsymbol{x}^0$, $\boldsymbol{z}^0$, $\boldsymbol{\mu}^0$ and $\boldsymbol{\beta}_k^0$ for partial observation
2: **for** $t = 0, 1, \ldots, T-1$ **do**
3:     $\boldsymbol{z}^{t+1} \leftarrow \mathcal{D}_{\eta/\rho}\left(\boldsymbol{x}^t - \boldsymbol{\mu}^t/\rho\right)$         ▷ Denoising step
4:     *// Complete observation*
5:     $\boldsymbol{x}^{t+1} \leftarrow \mathcal{F}^{-1}\left( \dfrac{\frac{2}{n}\sum_{k=1}^n \overline{\mathcal{F}(\boldsymbol{h}_k)}\mathcal{F}(\boldsymbol{y}_k) + \mathcal{F}\left(\rho \boldsymbol{z}^{t+1} + \boldsymbol{\mu}^t\right)}{\frac{2}{n}\sum_{k=1}^n |\mathcal{F}(\boldsymbol{h}_k)|^2 + \rho} \right)$
6:     *// Partial observation*
7:     $\boldsymbol{x}^{t+1} \leftarrow \mathcal{F}^{-1}\left( \dfrac{\sum_{k=1}^n \overline{\mathcal{F}(\boldsymbol{h}_k)}\mathcal{F}(\boldsymbol{r}_k^t + \boldsymbol{y}_k + \boldsymbol{\beta}_k^t/\rho) + \mathcal{F}\left(\boldsymbol{z}^{t+1} + \boldsymbol{\mu}^t/\rho\right)}{\sum_{k=1}^n |\mathcal{F}(\boldsymbol{h}_k)|^2 + 1} \right)$
8:     **for** $k = 1, \ldots, n$ **do**
9:        $\boldsymbol{r}_k^{t+1} \leftarrow \dfrac{\rho(\boldsymbol{h}_k * \boldsymbol{x}^{t+1} - \boldsymbol{y}_k) - \boldsymbol{\beta}_k^t}{\frac{2}{n}\boldsymbol{m}_k + \rho}$
10:       $\boldsymbol{\beta}_k^{t+1} \leftarrow \boldsymbol{\beta}_k^t + \rho\left(\boldsymbol{r}_k^{t+1} - (\boldsymbol{h}_k * \boldsymbol{x}^{t+1} - \boldsymbol{y}_k)\right)$
11:     **end for**
12:     $\boldsymbol{\mu}^{t+1} \leftarrow \boldsymbol{\mu}^t + \rho\left(\boldsymbol{z}^{t+1} - \boldsymbol{x}^{t+1}\right)$
13: **end for**
14: **return** $\boldsymbol{x}^T$

---

**Algorithm 4** `LocDeblur-TV` and `FedAvgDeblur-TV`

---

**Require:** Max iterations $T$, regularization parameter $\eta$, ADMM parameter $\rho$

1: Initialize $\boldsymbol{x}_k^0$, $\boldsymbol{z}_k^0$, $\boldsymbol{\mu}_k^0$ for $k = 1, \ldots, n$.
2: **for** $t = 0, 1, \ldots, T-1$ **do**
3:     **for** $k = 1, \ldots, n$ **in parallel do**
4:        $\boldsymbol{x}_k^t \leftarrow \bar{\boldsymbol{x}}^t$     *// Only for* `FedAvgDeblur`
5:        $\boldsymbol{z}_k^{t+1} \leftarrow \text{Shrink}_{\eta/\rho}\left(\boldsymbol{D}\boldsymbol{x}_k^t - \boldsymbol{\mu}_k^t/\rho\right)$     ▷ Soft shrinkage
6:        $\boldsymbol{x}_k^{t+1} \leftarrow \mathcal{F}^{-1}\left( \dfrac{\overline{\mathcal{F}(\boldsymbol{h}_k)}\mathcal{F}(\boldsymbol{y}_k) + \mathcal{F}\left(\rho \boldsymbol{D}^T(\boldsymbol{z}_k^{t+1} + \boldsymbol{\mu}_k^t/\rho)\right)}{|\mathcal{F}(\boldsymbol{h}_k)|^2 + \rho|\mathcal{F}(\boldsymbol{D})|^2} \right)$
7:        $\boldsymbol{\mu}_k^{t+1} \leftarrow \boldsymbol{\mu}_k^t + \rho\left(\boldsymbol{z}_k^{t+1} - \boldsymbol{D}\boldsymbol{x}_k^{t+1}\right)$
8:     **end for**
9:     $\bar{\boldsymbol{x}}^{t+1} \leftarrow \frac{1}{n}\sum_{k=1}^n \boldsymbol{x}_k^{t+1}$     *// Only for* `FedAvgDeblur`     ▷ Averaging step
                    ▷ Average each pixel only over observed clients under partial observation
10: **end for**
11: **return** $\bar{\boldsymbol{x}}^T$ *for* `FedAvgDeblur` or $\{\boldsymbol{x}_k^T\}$ *for* `LocDeblur`

---

observation data ($\boldsymbol{y}_k$), blur kernels ($\boldsymbol{h}_k$), and positional masks ($\boldsymbol{m}_k$). The algorithm directly solves the global optimization problem (2) or (1) using the classic ADMM framework.

The implementations for the TV and PnP variants are detailed in Algorithm 2 and Algorithm 3, respectively. For the TV variant, the algorithm uses variable splitting by introducing an auxiliary variable $\boldsymbol{z} = \boldsymbol{D}\boldsymbol{x}$ to handle the TV regularizer, and the solution to the $\boldsymbol{z}$-subproblem corresponds to a soft shrinkage operation. For the PnP variant, the variable splitting is simplified to $\boldsymbol{z} = \boldsymbol{x}$, where the update for $\boldsymbol{z}$ becomes a denoising step performed by the DRUNet model. In both variants, for the partial observation setting, the presence of the positional masks $\boldsymbol{m}_k$ requires introducing additional auxiliary variables $\boldsymbol{r}_k = \boldsymbol{h}_k * \boldsymbol{x} - \boldsymbol{y}_k$ to decouple the problem, similar to the approach used in the partial observation extension of `FedDeblur`.

**LocDeblur and FedAvgDeblur.** The core of both `LocDeblur` and `FedAvgDeblur` is a standard ADMM-based deblurring method that runs locally on each client. The implementation for the TV versions is unified in Algorithm 4, and for the PnP versions in Algorithm 5. The primary difference between the two versions lies in the local proximal update step: the TV variant uses a soft shrinkage operation for the auxiliary variable $\boldsymbol{z}_k$, while the PnP variant employs a denoising network.

---

**Algorithm 5** `LocDeblur-PnP` and `FedAvgDeblur-PnP`

---

**Require:** Max iterations $T$, regularization parameter $\eta$, ADMM parameter $\rho$
1: Initialize $\boldsymbol{x}_k^0, \boldsymbol{z}_k^0, \boldsymbol{\mu}_k^0$ for $k = 1, \dots, n$.
2: **for** $t = 0, 1, \dots, T-1$ **do**
3:     **for** $k = 1, \dots, n$ **in parallel do**
4:         $\boldsymbol{x}_k^t \leftarrow \bar{\boldsymbol{x}}^t$       // *Only for* `FedAvgDeblur`
5:         $\boldsymbol{z}_k^{t+1} \leftarrow \mathcal{D}_{\eta/\rho}\left(\boldsymbol{x}_k^t - \boldsymbol{\mu}_k^t/\rho\right)$       $\triangleright$ Denoising step
6:         $\boldsymbol{x}_k^{t+1} \leftarrow \mathcal{F}^{-1}\left(\frac{\overline{\mathcal{F}(\boldsymbol{h}_k)}\mathcal{F}(\boldsymbol{y}_k) + \mathcal{F}\left(\rho \boldsymbol{z}_k^{t+1} + \boldsymbol{\mu}_k^t\right)}{|\mathcal{F}(\boldsymbol{h}_k)|^2 + \rho}\right)$
7:         $\boldsymbol{\mu}_k^{t+1} \leftarrow \boldsymbol{\mu}_k^t + \rho\left(\boldsymbol{z}_k^{t+1} - \boldsymbol{x}_k^{t+1}\right)$
8:     **end for**
9:     $\bar{\boldsymbol{x}}^{t+1} \leftarrow \frac{1}{n}\sum_{k=1}^n \boldsymbol{x}_k^{t+1}$       // *Only for* `FedAvgDeblur`     $\triangleright$ Averaging step
                                $\triangleright$ Average each pixel only over observed clients under partial observation
10: **end for**
11: **return** $\bar{\boldsymbol{x}}^T$ *for* `FedAvgDeblur` or $\{\boldsymbol{x}_k^T\}$ *for* `LocDeblur`

---

`LocDeblur` is a fully non-collaborative baseline that represents a performance lower bound. Each client independently runs its local ADMM algorithm until convergence without any communication with the server or other clients. For evaluation purposes, the final restored images from all clients $\{\boldsymbol{x}_k^T\}$ are averaged.

`FedAvgDeblur` is inspired by the federated averaging (`FedAvg`) algorithm. Unlike `LocDeblur`, it involves communication in every round. In each round, every client begins with the global average model from the previous round, $\bar{\boldsymbol{x}}^t$, and performs a single-step local ADMM update to obtain its new local estimate, $\boldsymbol{x}_k^{t+1}$. The clients then upload these updated estimates to the server. The server computes their average to produce the new global model, $\bar{\boldsymbol{x}}^{t+1}$, and broadcasts it to all clients for the next round. Although `FedAvgDeblur` enables collaboration, it requires the direct transmission and averaging of the image estimates $\boldsymbol{x}_k$, which violates the "intermediate privacy" principle. Furthermore, since each client must independently perform the proximal update, it can be computationally expensive when using deep priors, potentially violating the "computational efficiency" principle.

**Modularity Comparison.** A key advantage of our `FedDeblur` framework (Algorithm 1) is its modularity, particularly when switching between different image priors. When changing from a TV regularizer to a PnP prior, the necessary modifications are confined entirely to the server-side update steps (line 7 versus lines 9 to 11). Crucially, the client-side computations and the communication protocol remain unchanged, making the switch completely transparent to the clients.

In contrast, the baseline methods require more substantial alterations. For instance, comparing `CenDeblur-TV` (Algorithm 2) and `CenDeblur-PnP` (Algorithm 3) reveals that nearly every step of the iterative process is modified due to the different variable splitting strategies. Similarly, for `LocDeblur` and `FedAvgDeblur`, switching from the TV version (Algorithm 4) to the PnP version (Algorithm 5) requires changing the entire local update logic for each client. For `FedAvgDeblur`, only the averaging step remains the same, while for `LocDeblur`, every step of the iteration is different. This inherent flexibility demonstrates the robustness and practical advantages of our proposed federated design.

# D   ADDITIONAL EXPERIMENTAL DETAILS

## D.1   IMPLEMENTATION DETAILS

Our experiments are conducted on a set of standard grayscale and color images, as shown in Figure 5. All images are resized to $256 \times 256$ pixels. We quantitatively evaluate the quality of the restored images using the peak signal-to-noise ratio (PSNR) and the structural similarity index (SSIM). Except for appendix D.4, the blurred observations in all experiments are corrupted by additive white

Gaussian noise with a standard deviation of 1. All iterative algorithms are run for a maximum of 500 iterations or until the relative change in the restored image falls below a threshold of $10^{-5}$, i.e., $\|\boldsymbol{x}^{t+1} - \boldsymbol{x}^t\|_2 / \|\boldsymbol{x}^t\|_2 < 10^{-5}$. For each method under each setting, hyperparameters are individually optimized via grid search. All experiments were implemented in Python using the PyTorch framework[1] and performed on a workstation equipped with an NVIDIA RTX 3090 GPU.

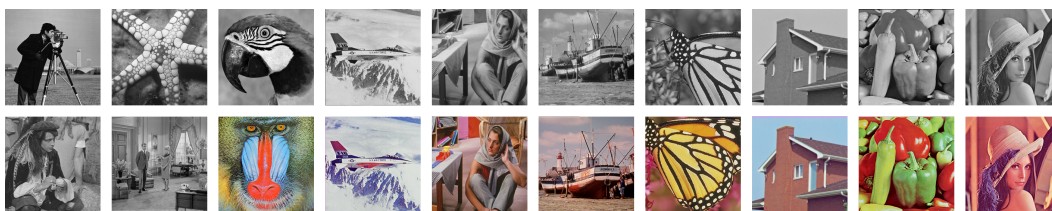

Figure 5: Test images used in our experiments. The grayscale set includes Cameraman, Fishstar, Parrot, Jetplane, Barbara, Boat, Butterfly, House, Pepper, Lena, Male, and Couple. The color set includes Baboon, Jetplane, Barbara, Boat, Butterfly, House, Pepper, and Lena.

For the complete observation setting, we simulate scenarios with $n$ clients, where $n$ ranges from 3 to 10. The blur kernel for each client is selected from a predefined set of 10 diverse kernels, including motion, Gaussian, disk, and synthesized types. The specific parameters for these 10 kernels are detailed in Table 3. An experiment with $n$ clients utilizes the first $n$ kernels from this table.

Table 3: Blur kernel parameters for each of the 10 clients in the complete observation setting.

| Client $k$ | 1 | 2 | 3 | 4 | 5 | 6 | 7 | 8 | 9 | 10 |
|---|---|---|---|---|---|---|---|---|---|---|
| Kernel Type | Motion | Gaussian | Synthetic | Motion | Disk | Motion | Gaussian | Disk | Synthetic | Motion |
| Parameters | len=15 $\theta=45°$ | size=9 $\sigma=2$ | $15 \times 15$ | len=20 $\theta=0°$ | r=5 | len=15 $\theta=135°$ | size=10 $\sigma=3$ | r=10 | $12 \times 12$ | len=20 $\theta=90°$ |

For the partial observation setting, the division of the scene into client-specific observation regions is described in section 4.1. We consider two distinct blur degradation scenarios: "motion" and "synthesis". In the "motion" scenario, each of the nine clients is assigned a unique motion blur kernel with varying lengths and angles, as specified in Table 4. In the "synthesis" scenario, the nine blur kernels are randomly generated camera-shake kernels. These kernels are generated using the public code from KAIR[2] with a fixed random seed to ensure reproducibility. The resulting kernels are visualized in Figure 2b.

Table 4: Motion blur kernel parameters for each of the 9 clients in the partial observation setting.

| Client $k$ | 1 | 2 | 3 | 4 | 5 | 6 | 7 | 8 | 9 |
|---|---|---|---|---|---|---|---|---|---|
| Length | 10 | 11 | 13 | 15 | 17 | 19 | 21 | 23 | 25 |
| Angle (°) | 0 | 22 | 45 | 67 | 90 | 112 | 135 | 157 | 180 |

## D.2 EFFECT OF THE NUMBER OF CLIENTS

To evaluate the scalability and effectiveness of our federated framework, we vary the number of clients $n$ from 1 to 10 in the complete observation setting. As shown in Figure 6, PSNR and SSIM increase steadily as more clients participate across different image priors and datasets. The largest gains occur when moving from a small number of clients, after which the curves gradually plateau, indicating that while adding more clients is always beneficial, the marginal utility diminishes. This consistent trend across different test data (grayscale and color) and priors (TV and PnP) confirms that the observed improvement is a fundamental property of our algorithm and the complementary information contributed by each client's unique degradation.

---

[1]Code is available at https://anonymous.4open.science/r/FedDeblur-53D7.
[2]https://github.com/cszn/KAIR/blob/master/utils/utils_deblur.py.

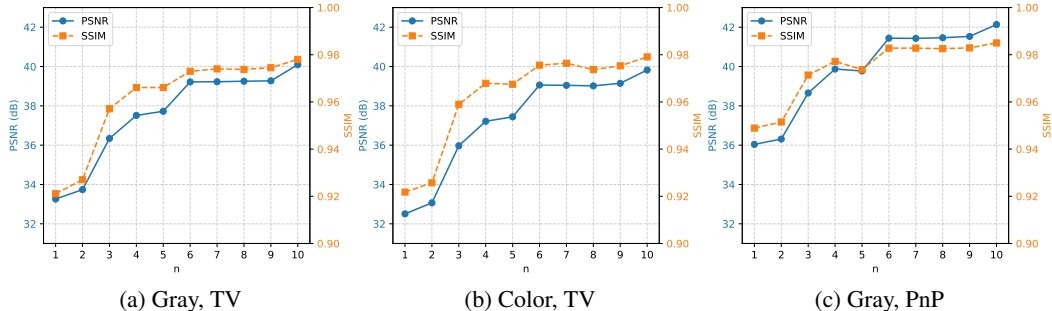

(a) Gray, TV         (b) Color, TV         (c) Gray, PnP

Figure 6: Performance of `FedDeblur` as the number of clients $n$ in the complete observation setting. Results are averaged over all test images for the TV prior on (a) grayscale and (b) color datasets, and for the PnP prior on (c) the grayscale dataset.

This trend arises from aggregating complementary information across clients. Each client provides an observation degraded by a unique blur kernel, which corrupts the underlying image in different ways. Figure 7 illustrates this effect: as $n$ increases from 3 to 7, the restored image recovers finer textures. The convergence plots in Figure 7d and 7e further show that more clients yield higher final quality and faster convergence.

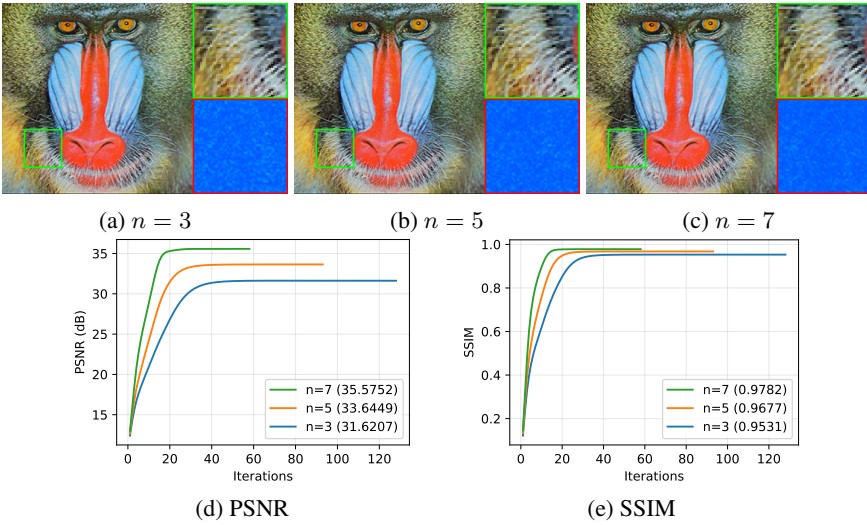

(a) $n = 3$         (b) $n = 5$         (c) $n = 7$

(d) PSNR                  (e) SSIM

Figure 7: Visual examples and convergence plots for different numbers of clients $n$ on the "Baboon" image with TV prior. (a-c) Reconstruction crops for $n = 3, 5, 7$. Each image is accompanied by an enlarged patch (top right) and its residual map (bottom right). PSNR and SSIM values are displayed on the top left of each image. (d-e) PSNR and SSIM versus iteration for varying $n$; final metric values are listed in the legends.

### D.3 PARTIAL CLIENT PARTICIPATION

In practical federated learning deployments, participating clients often consist of a wide range of devices with heterogeneous hardware, leading to varying local computation speeds (Li et al., 2020b; Ye et al., 2023). Furthermore, these devices are typically connected to diverse network environments with significant differences in bandwidth, latency, and stability. This can result in clients completing their communication at vastly different times, with some even failing to respond within a given time window (McMahan et al., 2017; Chen et al., 2021). These factors contribute to the "straggler" problem, where the server would be forced to wait for the slowest clients before performing a global aggregation, thereby reducing overall efficiency (Li et al., 2020b). A common strategy to mitigate

---

**Algorithm 6** `FedDeblur` for partial client participation

---

**Require:** Communication rounds $T$, regularization parameter $\eta$, ADMM parameter $\rho$
1: **Server**: Initialize $\boldsymbol{x}^0$, $\mathcal{S}_0 = \{1, \ldots, n\}$
2: **Clients** $k = 1, \ldots, n$: Initialize $\boldsymbol{x}_k^0, \boldsymbol{\lambda}_k^0, \boldsymbol{w}_k^0 = \boldsymbol{x}_k^0 + \boldsymbol{\lambda}_k^0/\rho$; and upload $\boldsymbol{w}_k^0$ to the server
3: **for** $t = 0, 1, \ldots, T-1$ **do**
4:     **server do**
5:         Aggregate $\boldsymbol{w}^t \leftarrow \frac{1}{|\mathcal{S}_t|} \sum_{k \in \mathcal{S}_t} \boldsymbol{w}_k^t$
6:         Server update same as Algorithm 1
7:         Select the active clients set $\mathcal{S}_{t+1}$ and broadcast $\boldsymbol{x}^{t+1}$ to the active clients
8:     **for each active client** $k \in \mathcal{S}_{t+1}$ **in parallel do**
9:         Receive global model $\boldsymbol{x}^{t+1}$ from the server
10:        Update local variables same as Algorithm 1
11:        Compute $\boldsymbol{w}_k^{t+1}$ and upload to the server
12:     **end for**
13: **end for**
14: **return** global estimate $\boldsymbol{x}^T$

---

this is to proceed with aggregation after a sufficient number of clients have responded, which requires the federated optimization algorithm to support partial client participation.

We model this scenario by having the server select a random subset of active clients $\mathcal{S}^t \subseteq \{1, 2, \ldots, n\}$ in each communication round $t$. The modified workflow is detailed in Algorithm 6. To study the impact of partial participation, we conduct experiments on a system with a total of $n = 10$ clients, varying the number of randomly selected active clients, $p = |\mathcal{S}^t|$, from 2 to 10 in each round.

The results, shown in Figure 8, demonstrate that while a higher participation rate consistently improves performance, our framework remains robust and effective even at low participation rates. As shown in the convergence plots in Figure 9, lower participation rates lead to slower convergence and more pronounced oscillations in the PSNR curve. This increased variability is expected, as each server update is based on a smaller, more random subset of client information, making the global update trajectory more stochastic. Nonetheless, the visual results confirm that high-quality restoration is achievable even when only a small fraction of clients participates in each round.

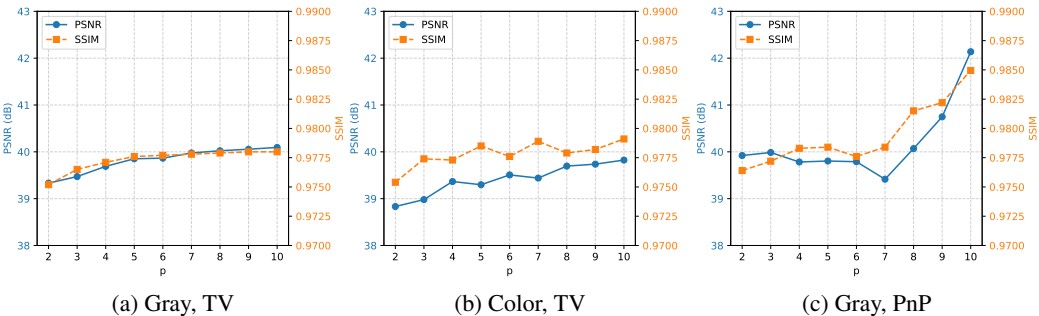

     (a) Gray, TV           (b) Color, TV           (c) Gray, PnP

Figure 8: Performance of `FedDeblur` with partial client participation on a system with $n = 10$ total clients, as the number of active clients per round $p$ varied from 2 to 10. Results are shown for (a) grayscale images with TV prior, (b) color images with TV prior, and (c) grayscale images with PnP prior.

### D.4 ROBUSTNESS TO OBSERVATION NOISE

To assess the robustness of our framework to observation noise, we evaluate performance under additive white Gaussian noise with standard deviation $\sigma$ ranging from 0 (noiseless) to 3. Complete quantitative results are reported in Table 5 (results for $\sigma = 1$ appear in Table 1 in section 4.2), and Figure 10 summarizes the case of $n = 10$ color images.

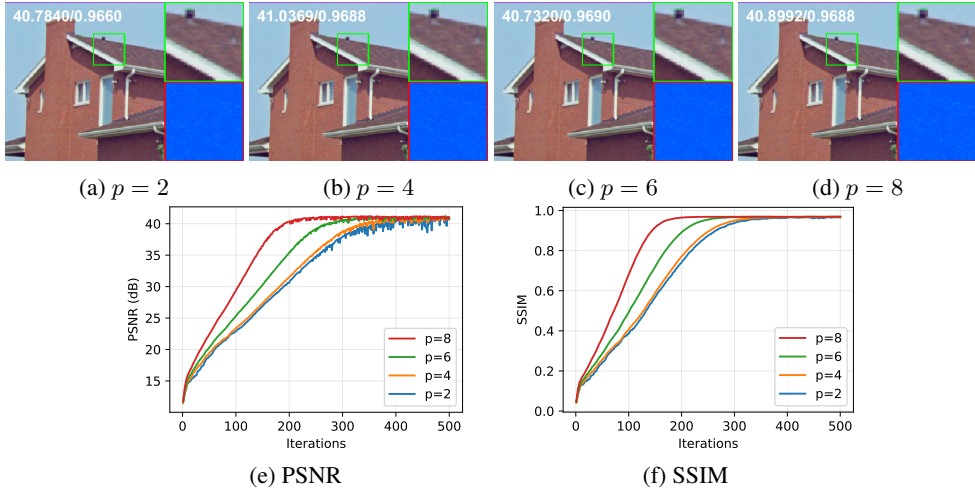

Figure 9: Visual results and convergence analysis for partial client participation on the House image with $n = 10$ total clients and the TV prior. (a-d) Final deblurring results for different numbers of active clients $p$. Each image is accompanied by an enlarged patch (top right) and its residual map (bottom right). PSNR and SSIM values are displayed on the top left of each image. (e-f) Corresponding convergence curves for PSNR and SSIM.

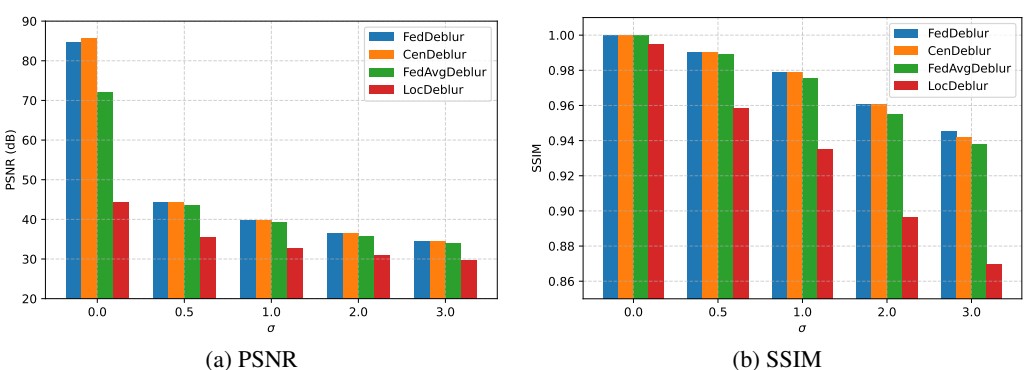

Figure 10: Performance under varying noise levels ($\sigma$) in the complete observation setting with $n = 10$ on color images.

Across all noise levels, the relative ranking of methods remains stable, confirming that the advantages of our federated design are fundamental and not tied to a specific noise setting. As noise increases, all methods degrade as expected, yet `FedDeblur` closely matches the idealized centralized oracle `CenDeblur` with only a small gap. This shows strong robustness of our framework to noise. Our consensus mechanism effectively aggregates out the effects of random noise from multiple observations, yielding a cleaner global reconstruction than non-collaborative or naive averaging approaches.

In particular, as shown in Figure 10b, the SSIM gap between the collaborative methods (`FedDeblur`, `CenDeblur`, `FedAvgDeblur`) and the non-collaborative baseline (`LocDeblur`) grows with noise, which highlights the importance of information fusion in noisy scenarios. The consistent margin between `FedDeblur` and `FedAvgDeblur` further supports the advantage of our consensus optimization over naive parameter averaging, especially when observations are heavily corrupted. Even in the noiseless case ($\sigma = 0$), where PSNR values are very high, the architectural advantages of our method remain evident.

Table 5: Comparison with TV regularization across different noise levels ($\sigma$). For each metric, the best result is highlighted in magenta and the second-best in cyan .

| Setup | Type | FedDeblur PSNR | FedDeblur SSIM | CenDeblur PSNR | CenDeblur SSIM | FedAvgDeblur PSNR | FedAvgDeblur SSIM | LocDeblur PSNR | LocDeblur SSIM |
|---|---|---|---|---|---|---|---|---|---|
| \multicolumn{10}{c}{$\sigma = 0$} |||||||||
| $n = 10$ | Gray | 81.3782 | 1.0000 | 81.2093 | 1.0000 | 68.0343 | 0.9999 | 44.9688 | 0.9944 |
| | Color | 84.6834 | 1.0000 | 85.5895 | 1.0000 | 71.8875 | 1.0000 | 44.2351 | 0.9947 |
| $n = 5$ | Gray | 74.3669 | 1.0000 | 74.5099 | 1.0000 | 65.0938 | 0.9999 | 46.2258 | 0.9960 |
| | Color | 76.6074 | 1.0000 | 76.6508 | 1.0000 | 66.8386 | 0.9999 | 45.2914 | 0.9961 |
| $n = 3$ | Gray | 64.2039 | 0.9999 | 64.1226 | 0.9999 | 58.0734 | 0.9994 | 42.7921 | 0.9922 |
| | Color | 66.4204 | 0.9999 | 66.2443 | 0.9999 | 60.7934 | 0.9997 | 41.6485 | 0.9919 |
| Motion | Gray | 41.4498 | 0.9866 | 41.7750 | 0.9865 | 23.2112 | 0.7302 | 23.2817 | 0.7364 |
| | Color | 40.7575 | 0.9871 | 41.0725 | 0.9872 | 22.5916 | 0.7077 | 22.6414 | 0.7086 |
| Synthesis | Gray | 35.6857 | 0.9634 | 35.5461 | 0.9686 | 21.6630 | 0.6642 | 21.5664 | 0.6661 |
| | Color | 34.5416 | 0.9618 | 34.0416 | 0.9645 | 21.3235 | 0.6587 | 21.2558 | 0.6509 |
| \multicolumn{10}{c}{$\sigma = 0.5$} |||||||||
| $n = 10$ | Gray | 44.3863 | 0.9898 | 44.3857 | 0.9899 | 43.7826 | 0.9881 | 36.1205 | 0.9634 |
| | Color | 44.1880 | 0.9901 | 44.1937 | 0.9902 | 43.5805 | 0.9889 | 35.3630 | 0.9583 |
| $n = 5$ | Gray | 41.3922 | 0.9818 | 41.3976 | 0.9818 | 41.1850 | 0.9811 | 36.4778 | 0.9609 |
| | Color | 41.1870 | 0.9838 | 41.1890 | 0.9838 | 40.9501 | 0.9822 | 35.6259 | 0.9599 |
| $n = 3$ | Gray | 39.5199 | 0.9763 | 39.5249 | 0.9764 | 39.4486 | 0.9754 | 35.2012 | 0.9549 |
| | Color | 39.2602 | 0.9771 | 39.2200 | 0.9777 | 39.1419 | 0.9757 | 34.3165 | 0.9544 |
| Motion | Gray | 36.6886 | 0.9594 | 36.6801 | 0.9594 | 23.2099 | 0.7301 | 23.2805 | 0.7362 |
| | Color | 35.9757 | 0.9585 | 35.8457 | 0.9589 | 22.4477 | 0.7169 | 22.6406 | 0.7085 |
| Synthesis | Gray | 33.4607 | 0.9372 | 33.6281 | 0.9370 | 21.6621 | 0.6641 | 21.5366 | 0.6658 |
| | Color | 32.5743 | 0.9353 | 32.7310 | 0.9357 | 21.3575 | 0.6530 | 21.2060 | 0.6539 |
| \multicolumn{10}{c}{$\sigma = 2$} |||||||||
| $n = 10$ | Gray | 36.7494 | 0.9595 | 36.7347 | 0.9597 | 36.3046 | 0.9546 | 31.5904 | 0.9132 |
| | Color | 36.3651 | 0.9605 | 36.3690 | 0.9604 | 35.7339 | 0.9549 | 30.7809 | 0.8961 |
| $n = 5$ | Gray | 34.6002 | 0.9420 | 34.6011 | 0.9420 | 34.1911 | 0.9413 | 31.7884 | 0.9142 |
| | Color | 34.0418 | 0.9448 | 34.0465 | 0.9447 | 33.9568 | 0.9404 | 30.9929 | 0.9043 |
| $n = 3$ | Gray | 33.4663 | 0.9336 | 33.4664 | 0.9336 | 33.3500 | 0.9322 | 31.5310 | 0.9113 |
| | Color | 33.0918 | 0.9303 | 33.0994 | 0.9303 | 32.9590 | 0.9290 | 30.7611 | 0.8956 |
| Motion | Gray | 31.0227 | 0.8958 | 30.9980 | 0.8962 | 23.1970 | 0.7290 | 23.2635 | 0.7348 |
| | Color | 30.2842 | 0.8881 | 30.2700 | 0.8870 | 22.4907 | 0.7160 | 22.6069 | 0.7097 |
| Synthesis | Gray | 29.7694 | 0.8773 | 29.7735 | 0.8769 | 21.6534 | 0.6632 | 21.4859 | 0.6636 |
| | Color | 28.9029 | 0.8659 | 28.9224 | 0.8660 | 21.3164 | 0.6580 | 21.2428 | 0.6502 |
| \multicolumn{10}{c}{$\sigma = 3$} |||||||||
| $n = 10$ | Gray | 34.9053 | 0.9399 | 34.9918 | 0.9455 | 34.3851 | 0.9388 | 30.3957 | 0.8923 |
| | Color | 34.4920 | 0.9453 | 34.4931 | 0.9418 | 33.8898 | 0.9379 | 29.5742 | 0.8697 |
| $n = 5$ | Gray | 33.0265 | 0.9255 | 33.0331 | 0.9254 | 32.3920 | 0.9230 | 30.6704 | 0.8776 |
| | Color | 32.5696 | 0.9258 | 32.5790 | 0.9256 | 32.2623 | 0.9239 | 29.7825 | 0.8814 |
| $n = 3$ | Gray | 31.7953 | 0.9143 | 32.1334 | 0.9137 | 31.4859 | 0.9099 | 30.4770 | 0.8924 |
| | Color | 31.5071 | 0.9125 | 31.5205 | 0.9125 | 31.2762 | 0.9102 | 29.5814 | 0.8819 |
| Motion | Gray | 29.6523 | 0.8708 | 29.7025 | 0.8694 | 23.1817 | 0.7278 | 23.2491 | 0.7330 |
| | Color | 28.9283 | 0.8605 | 28.9197 | 0.8565 | 22.4484 | 0.7164 | 22.5858 | 0.7075 |
| Synthesis | Gray | 28.6313 | 0.8529 | 28.6659 | 0.8534 | 21.6431 | 0.6621 | 21.4184 | 0.6654 |
| | Color | 27.9674 | 0.8371 | 27.9135 | 0.8386 | 21.0973 | 0.6616 | 21.2364 | 0.6493 |

## D.5 Numerical Convergence Analysis

To better illustrate the optimization dynamics of our framework, we present a numerical analysis of convergence for several representative settings, shown in Figure 11 to 13.

The federated methods, `FedDeblur` and `FedAvgDeblur` begin with low metric values because the server uses random initialization to comply with privacy principles that prohibit access to any client data. By contrast, the centralized method, `CenDeblur`, is initialized with the average of all blurry observations, and the non-collaborative baseline, `LocDeblur`, starts from each client's observation, giving both a significant head start. Despite the lower starting point, `FedDeblur` improves rapidly, and its trajectory is nearly parallel to that of the idealized `CenDeblur` oracle, indicating a comparable convergence rate. Moreover, `FedDeblur` consistently converges faster and to a better result than the naive federated baseline, `FedAvgDeblur`. This holds for both TV and PnP priors, highlighting the effectiveness of the proposed consensus mechanism.

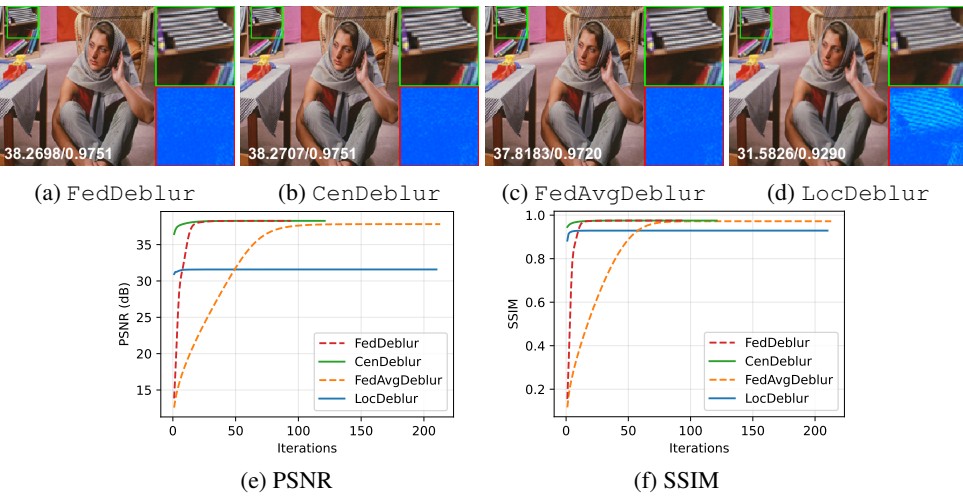

Figure 11: Visual results and convergence analysis for the complete observation setting with $n = 10$ clients on the color Barbara image using the TV prior. (a-d) Final results; Each image is accompanied by an enlarged patch (top right) and its residual map (bottom right); PSNR and SSIM are shown in the bottom left of each image. (e-f) PSNR and SSIM versus iteration number.

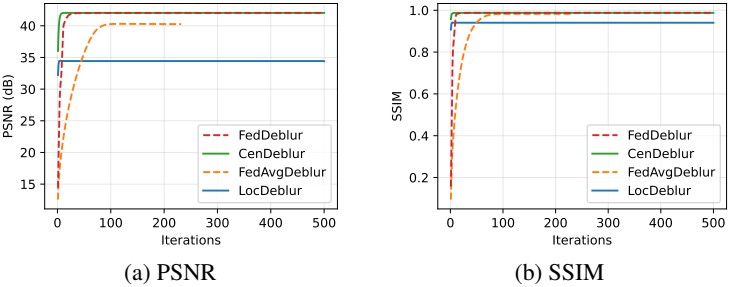

Figure 12: Convergence analysis for the complete observation setting with $n = 10$ clients on the grayscale Barbara image using the PnP prior. The corresponding visual results appear in Figure 3.

Furthermore, in the partial observation setting (Figure 13), the PSNR curve for `FedDeblur` exhibits periodic fluctuations. To study this behavior, we track several quantities: the consensus error $\varepsilon^t = x_k^t - x^t$, the change in the global primal variable $\Delta x^t = x^t - x^{t-1}$, the change in the shared variable $\Delta w^t = w^t - w^{t-1}$, and the change in PSNR per iteration. As shown in Figure 13g, these metrics are strongly correlated and oscillatory.

This behavior reflects a feedback loop caused by the lag between local optimization and global consensus. In each iteration, clients update $x_k$ to balance fidelity to private data with agreement to the global model from the *previous* round. In overlapping regions, different blur kernels yield conflicting estimates for the same pixels. The server aggregates these into a new global model that can be a compromise and thus suboptimal for any individual client. The induced consensus error is stored in the dual variables, which act as memory. In the next iteration, these dual variables drive a strong correction toward consensus; this can overshoot and produce oscillations across iterations.

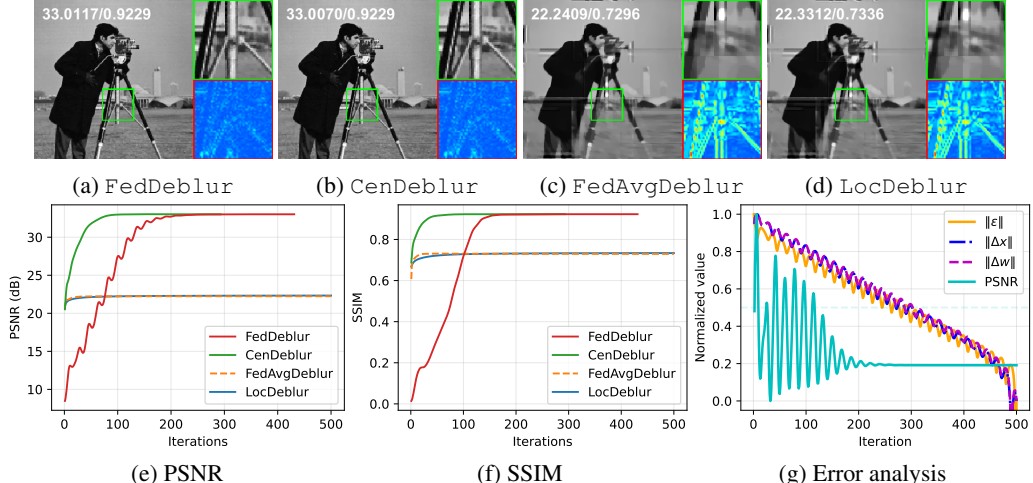

(a) `FedDeblur`  (b) `CenDeblur`  (c) `FedAvgDeblur`  (d) `LocDeblur`

(e) PSNR  (f) SSIM  (g) Error analysis

Figure 13: Visual results and convergence analysis for the partial observation setting with motion blur on the Cameraman image using the TV prior. (a-d) Final results; Each image is accompanied by an enlarged patch (top right) and its residual map (bottom right); PSNR and SSIM are shown in the top left of each image. (e-f) Performance metrics versus iteration. (g) The consensus error ($\|\boldsymbol{\varepsilon}^t\|$), changes in the local and shared variables ($\|\Delta\boldsymbol{x}^t\|$, $\|\Delta\boldsymbol{w}^t\|$), and the change in PSNR. Data in (g) were log-transformed and linearly scaled for clarity.

The synchronized peaks in the error metrics and the PSNR derivative in Figure 13g support this explanation: a build-up of consensus error triggers system-wide corrections, which manifest as periodic fluctuations.

To further demonstrate the superior convergence properties of our ADMM-based federated optimization framework, we compare it against a typical gradient descent (GD) based strategy, which serves as the backbone for many federated learning algorithms (e.g. McMahan et al. (2017); Karimireddy et al. (2020)). We formulate the `FedGDDeblur` baseline by directly applying gradient descent to the global objective

$$\min_{\boldsymbol{x}} \quad \frac{1}{n}\sum_{k=1}^{n}\|\boldsymbol{h}_k * \boldsymbol{x} - \boldsymbol{y}_k\|_2^2 + \eta\|\boldsymbol{D}\boldsymbol{x}\|_1.$$

The update rule for gradient descent with a learning rate $\alpha$ is given by

$$\boldsymbol{x}^{t+1} = \boldsymbol{x}^t - \alpha\left[\frac{2}{n}\sum_{k=1}^{n}\overline{\boldsymbol{h}}_k * \left(\boldsymbol{h}_k * \boldsymbol{x}^t - \boldsymbol{y}_k\right) - \eta\operatorname{div}\left(\frac{\boldsymbol{D}\boldsymbol{x}^t}{|\boldsymbol{D}\boldsymbol{x}^t|}\right)\right],$$

where $\overline{\boldsymbol{h}}_k$ denotes the flipped kernel (adjoint of convolution), and the regularization gradient is derived from the smoothed total variation. We implement this in a federated manner as shown in Algorithm 7, where clients compute gradients of the data fidelity term locally, and the server aggregates them and applies the regularization gradient.

We evaluated both methods on the color "Jetplane" image under the complete observation setting ($n = 10$) with TV regularization. As shown in Figure 14, `FedDeblur` significantly outperforms `FedGDDeblur` in both convergence speed and restoration quality. The visual results in Figure 14a and 14b reveal that `FedGDDeblur` fails to effectively remove blur and noise, resulting in artifacts, whereas `FedDeblur` produces a sharp, clean image.

The quantitative curves in Figure 14c and 14d confirm that `FedDeblur` achieves high PSNR and SSIM values within very few iterations. In contrast, `FedGDDeblur` converges extremely slowly and plateaus at a much lower quality level. Furthermore, the objective function plot in Figure 14e shows that our ADMM-based approach minimizes the energy function rapidly, while the GD-based approach struggles with the ill-conditioned nature of the deblurring problem. This comparison underscores the advantage of using proximal splitting methods (which handle non-smooth regularizers and ill-posed operators robustly) over standard gradient descent in federated inverse problems.

---

**Algorithm 7** `FedGDDeblur`

---

**Require:** Communication rounds $T$, learning rate $\alpha$, regularization parameter $\eta$
1: **Server**: Initialize global estimate $\boldsymbol{x}^0$
2: **for** $t = 0, 1, \ldots, T-1$ **do**
3:      **for each client** $k = 1, \ldots, n$ **in parallel do**
4:          Receive global model $\boldsymbol{x}^t$ from the server
5:          $\boldsymbol{g}_k^t \leftarrow 2\bar{\boldsymbol{h}}_k * (\boldsymbol{h}_k * \boldsymbol{x}^t - \boldsymbol{y}_k)$                             ▷ Compute local gradient
6:          Upload $\boldsymbol{g}_k^t$ to the server
7:      **end for**
8:      **Server do**
9:          $\bar{\boldsymbol{g}}_{\text{data}}^t \leftarrow \frac{1}{n} \sum_{k=1}^n \boldsymbol{g}_k^t$                        ▷ Aggregate data fidelity gradients
10:         $\boldsymbol{g}_{\text{reg}}^t \leftarrow -\eta \operatorname{div}\left(\frac{\boldsymbol{D}\boldsymbol{x}^t}{|\boldsymbol{D}\boldsymbol{x}^t|}\right)$             ▷ Compute regularization gradient
11:         $\boldsymbol{x}^{t+1} \leftarrow \boldsymbol{x}^t - \alpha\left(\bar{\boldsymbol{g}}_{\text{data}}^t + \boldsymbol{g}_{\text{reg}}^t\right)$           ▷ Update global estimate
12: **end for**
13: **return** final estimate $\boldsymbol{x}^T$

---

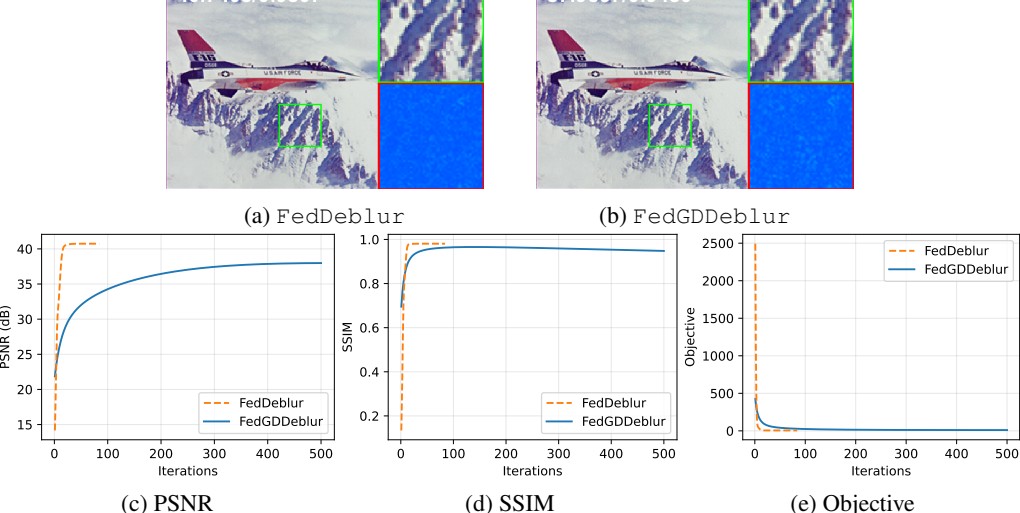

(a) `FedDeblur`                        (b) `FedGDDeblur`

(c) PSNR               (d) SSIM              (e) Objective

Figure 14: Comparison between `FedDeblur` and the gradient descent baseline `FedGDDeblur` on the color "Jetplane" image ($n = 10$, complete observation, TV prior). (a-b) Visual results with PSNR and SSIM shown in the top left of each image. (c-e) Convergence curves for PSNR, SSIM, and the objective function value.

### D.6   COMPUTATIONAL EFFICIENCY AND RUNTIME ANALYSIS

To further evaluate the computational resource consumption of the proposed framework, we recorded the average wall-clock time per communication round and the total time required to reach specific image quality thresholds. We compared the two federated methods, `FedDeblur` and `FedAvgDeblur`, in the complete observation setting with $n = 10$ clients using TV regularization. The experiments were performed on the same hardware described in appendix D.1. Table 6 summarizes the results. The first two columns report the average time per iteration averaged over the entire grayscale and color datasets, respectively. The remaining columns detail the number of communication rounds and the total cumulative time required to reach specific PSNR (35, 37, 38 dB) and SSIM (0.95, 0.97, 0.975) targets on the color "Barbara" image.

The results demonstrate the significant efficiency advantage of our method. In terms of per-iteration cost, `FedDeblur` is approximately $2\times$ to $3\times$ faster than `FedAvgDeblur`. Furthermore, `FedDeblur` exhibits a much faster convergence rate. To achieve a PSNR of 37 dB, our method requires only 14 rounds (0.37 seconds), whereas `FedAvgDeblur` requires 82 rounds (4.35 sec-

Table 6: Runtime and convergence speed comparison between `FedDeblur` and `FedAvgDeblur`. The average iteration time is computed over the entire dataset. The convergence targets (rounds / total time) are reported for the color "Barbara" image ($n = 10$, complete observation, TV prior). Entries marked with "−" indicate that the method failed to reach the target within 500 rounds.

| Method | Avg. Time per Round (s) | | Rounds / Total Time (s) to reach target | | | | | |
|---|---|---|---|---|---|---|---|---|
| | Gray | Color | PSNR (dB) | | | SSIM | | |
| | | | 35 | 37 | 38 | 0.95 | 0.97 | 0.975 |
| FedDeblur | **0.0129** | **0.0275** | **12 / 0.32** | **14 / 0.37** | **20 / 0.53** | **12 / 0.32** | **15 / 0.40** | **42 / 1.11** |
| FedAvgDeblur | 0.0370 | 0.0560 | 64 / 3.41 | 82 / 4.35 | − | 65 / 3.46 | 84 / 4.45 | − |

onds)—a speedup of over $10\times$ in total time. More importantly, `FedDeblur` is able to reach higher quality thresholds (PSNR 38 dB and SSIM 0.975) that `FedAvgDeblur` fails to attain even after extended iterations. This confirms that the consensus-based optimization of `FedDeblur` not only converges faster but also converges to a superior solution compared to the naive averaging of local estimates.

### D.7 SCALABILITY TO HIGH-RESOLUTION AND REAL-WORLD SCENARIOS

To further validate the scalability and robustness of our framework in more realistic and challenging environments, we extended our evaluation to high-resolution data and real-world multi-view scenarios.

First, we assessed the performance on high-definition images using the DIV2K dataset (Agustsson & Timofte, 2017). We selected real-world images with an resolution of $2040 \times 1356$. To simulate a large-scale federated system, we constructed a scenario with $n = 16$ clients arranged in a $4 \times 4$ grid, where each client captures a high-resolution partial view of $600 \times 500$ pixels. This setup represents a significant increase in scale compared to previous experiments. Furthermore, to heighten the difficulty of the restoration task, we employed larger $25 \times 25$ synthetic blur kernels. A larger kernel size induces more severe blur and stronger inter-pixel dependencies, making the inverse problem significantly more ill-posed. The restoration was performed using the PnP prior following the same protocol described in appendix D.1.

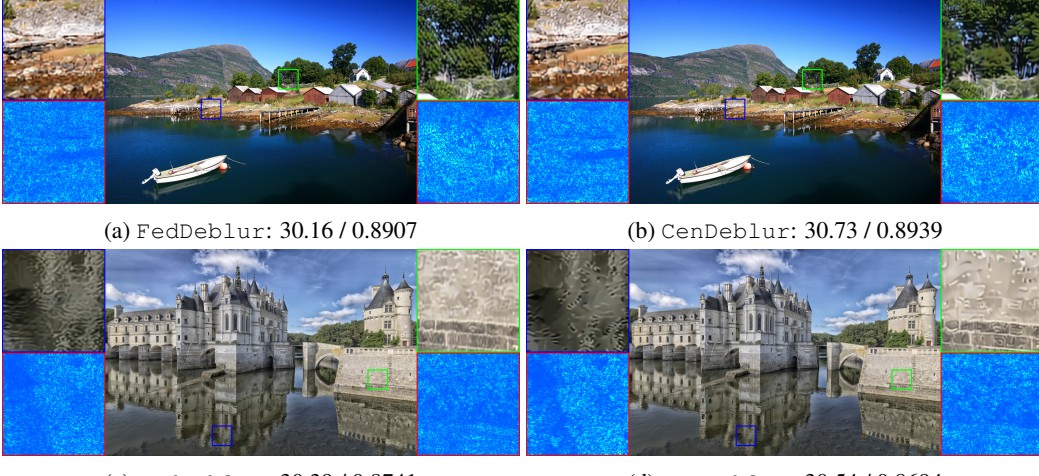

(a) `FedDeblur`: 30.16 / 0.8907          (b) `CenDeblur`: 30.73 / 0.8939

(c) `FedDeblur`: 30.39 / 0.8741          (d) `CenDeblur`: 30.54 / 0.8694

Figure 15: Visual comparison on high-resolution images from the DIV2K dataset. The scene is captured by $n = 16$ clients arranged in a $4 \times 4$ grid with partial overlap, degraded by $25 \times 25$ synthetic blur kernels. PSNR (dB) and SSIM are reported below each image.

As illustrated in Figure 15, `FedDeblur` maintains exceptional performance even in this challenging setting. Quantitatively, the PSNR and SSIM scores are comparable to those of the idealized centralized baseline (`CenDeblur`). Qualitatively, the restored images exhibit sharp details, and

notably, there are no visible seam artifacts at the boundaries of the overlapping client sub-regions. This confirms that our consensus mechanism effectively fuses local information into a coherent global image, scaling seamlessly to larger client arrays and higher resolutions.

Subsequently, we evaluated our method in a more realistic multi-view scenario using the DTU MVS dataset (Aanæs et al., 2016). Unlike the synthetic cropping used in previous experiments, we selected five adjacent views of a single scene to represent five independent clients. These images naturally exhibit distinct perspectives and varying lighting conditions, thereby introducing significant data heterogeneity that better reflects practical federated deblurring deployments.

To mitigate the domain gap caused by physical camera displacement and sensor variations, we performed standard pre-processing steps prior to optimization: the enhanced correlation coefficient (ECC) algorithm (Evangelidis & Psarakis, 2008) was used for spatial alignment, and histogram matching was applied for color correction. It is important to note that these pre-processing steps are external to our `FedDeblur` algorithm.

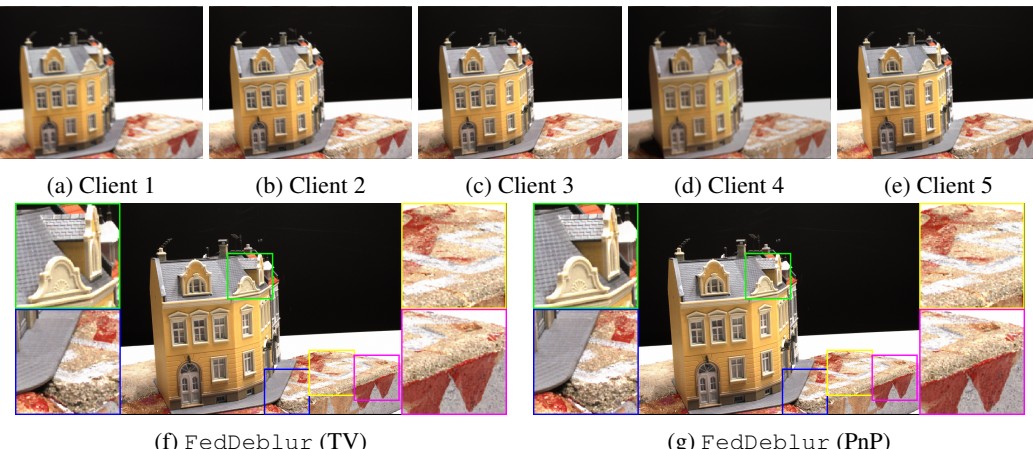

(a) Client 1      (b) Client 2      (c) Client 3      (d) Client 4      (e) Client 5

(f) `FedDeblur` (TV)             (g) `FedDeblur` (PnP)

Figure 16: Visual evaluation on real-world multi-view data from the DTU MVS dataset. The system consists of $n = 5$ clients, each capturing the scene from a different viewing angle with varying lighting conditions. (a-e) The blurred observations held by each client after preprocessing. (f-g) The global clear image recovered by `FedDeblur` using TV and PnP priors, respectively.

Since ground truth is unavailable for this specific real-world configuration, we rely on visual inspection. As shown in Figure 16, the input observations from the five clients suffer from significant degradation. However, `FedDeblur` successfully integrates the information from these heterogeneous views. Both the TV and PnP variants recover sharp geometric structures and fine textures (e.g. details of roof tiles and rough surfaces) while effectively suppressing noise. This demonstrates that our consensus-based framework is robust not only to synthetic partitions but also to the geometric and photometric inconsistencies inherent in real-world multi-view imaging.

### D.8 SENSITIVITY TO HYPERPARAMETERS

We examine the sensitivity of `FedDeblur` to its two hyperparameters: the regularization parameter $\eta$, which balances the data fidelity and prior terms, and the ADMM penalty parameter $\rho$, which sets the strength of the constraint penalty. We evaluate performance on an $11 \times 11$ grid centered at the grid-searched optimum, with adjacent values differing by $10\%$. Figure 17 shows performance surfaces for two representative cases: complete observation with $n = 5$ and partial observation with synthesized kernels, both using a TV prior.

The results show that the framework is robust to both hyperparameters. In particular, performance is very stable with respect to $\rho$. As shown across all subplots in Figure 17, the surfaces are nearly flat along the $\rho$-axis, indicating that the final restoration quality is largely insensitive to $\rho$ over a wide range. The convergence analysis in Figure 18 provides a more dynamic perspective. As shown in Figure 18m and 18n, while different values of $\rho$ influence the convergence path and speed, they

all lead to nearly identical final performance. This confirms that while $\rho$ can be tuned to affect convergence speed, its impact on the final restoration quality is minimal.

As expected, the regularization parameter $\eta$ has a more pronounced effect. Even so, the landscape is smooth with a broad peak around the optimum, without sharp drops for nearby values. The convergence curves in Figure 18k and 18l further clarify this relationship. Different values of $\eta$ lead to different final performance levels, as expected, since this parameter controls the trade-off between data fidelity and regularization. However, the visual results and the convergence plots show that performance degrades gracefully away from the optimal value rather than collapsing. This reinforces the finding from the surface plots: while $\eta$ matters, a near-optimal value suffices for strong results, and the method is not overly sensitive to its precise setting.

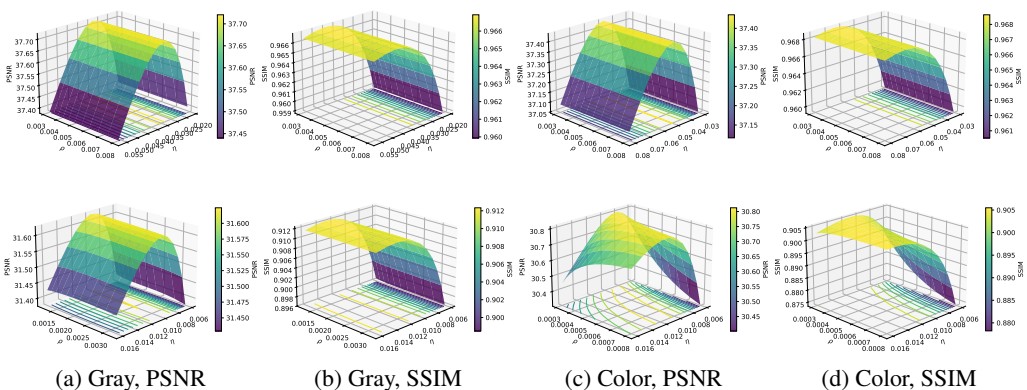

(a) Gray, PSNR      (b) Gray, SSIM      (c) Color, PSNR      (d) Color, SSIM

Figure 17: Sensitivity of `FedDeblur` to the regularization parameter $\eta$ and the ADMM penalty parameter $\rho$. First row: complete observation with $n = 5$. Second row: partial observation with synthesized kernels.

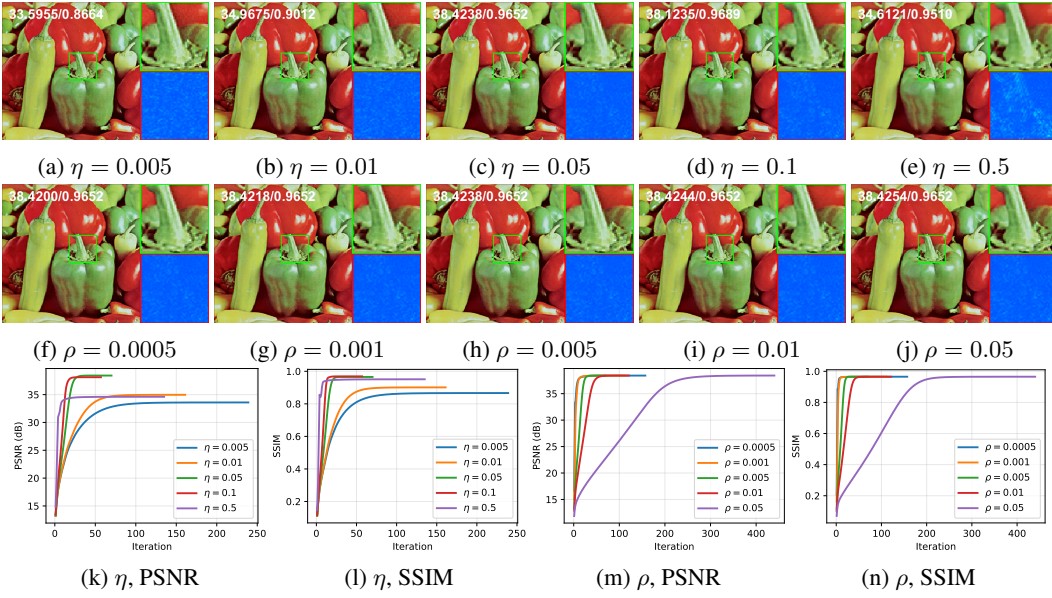

(a) $\eta = 0.005$    (b) $\eta = 0.01$    (c) $\eta = 0.05$    (d) $\eta = 0.1$    (e) $\eta = 0.5$

(f) $\rho = 0.0005$    (g) $\rho = 0.001$    (h) $\rho = 0.005$    (i) $\rho = 0.01$    (j) $\rho = 0.05$

(k) $\eta$, PSNR      (l) $\eta$, SSIM      (m) $\rho$, PSNR      (n) $\rho$, SSIM

Figure 18: Visual and numerical analysis of hyperparameter sensitivity for the complete observation setting with $n = 5$ on the Pepper image using the TV prior. (a-e) Deblurring results for different values of $\eta$ with a fixed $\rho = 0.005$. (f-j) Deblurring results for different values of $\rho$ with a fixed $\eta = 0.05$. Each image is accompanied by an enlarged patch (top right) and its residual map (bottom right). PSNR and SSIM values are displayed on the top left of each image. (k-l) Convergence curves showing the effect of varying $\eta$. (m-n) Convergence curves showing the effect of varying $\rho$.

# E  THEORETICAL CONVERGENCE ANALYSIS

## E.1  ASSUMPTIONS AND PRELIMINARIES

To facilitate the analysis, we denote

$$f_k(\boldsymbol{x}_k) = \frac{1}{n} \|\boldsymbol{H}_k \boldsymbol{x}_k - \boldsymbol{y}_k\|_2^2, \ k = 1, \ldots, n, \quad \text{and} \quad g(\boldsymbol{x}) = \eta R(\boldsymbol{x}), \tag{11}$$

where $\boldsymbol{H}_k$ is the linear operator corresponding to the convolution with kernel $\boldsymbol{h}_k$. We also define the original objective function from (2) as

$$P(\boldsymbol{x}) = \sum_{k=1}^n f_k(\boldsymbol{x}) + g(\boldsymbol{x}).$$

We begin by formally stating our assumptions and then present basic properties of the objective function.

**Assumption 1.** *The regularization function $R : \mathbb{R}^d \to \mathbb{R} \cup \{+\infty\}$ is proper, lower semi-continuous, convex, and bounded below.*

Note that $R$ here can be non-smooth, such as the $\ell_1$ norm. For implicit regularization, such as a PnP prior, if the denoiser satisfies appropriate conditions, it corresponds to the proximal operator of a convex function (Gribonval & Nikolova, 2020; Wei et al., 2025). In that case, Assumption 1 holds.

**Property 1.** *Each local fidelity function $f_k : \mathbb{R}^d \to \mathbb{R}$ is convex and has an L-Lipschitz continuous gradient (L-smooth); that is, there exists $L > 0$ such that for any $\boldsymbol{x}_1, \boldsymbol{x}_2 \in \mathbb{R}^d$,*

$$\|\nabla f_k(\boldsymbol{x}_1) - \nabla f_k(\boldsymbol{x}_2)\|_2 \leqslant L\|\boldsymbol{x}_1 - \boldsymbol{x}_2\|_2.$$

*Proof.* The gradient and Hessian of $f_k(\boldsymbol{x})$ are

$$\nabla f_k(\boldsymbol{x}) = \frac{2}{n} \boldsymbol{H}_k^T (\boldsymbol{H}_k \boldsymbol{x} - \boldsymbol{y}_k) \quad \text{and} \quad \nabla^2 f_k(\boldsymbol{x}) = \frac{2}{n} \boldsymbol{H}_k^T \boldsymbol{H}_k.$$

Since the Hessian $\nabla^2 f_k(\boldsymbol{x})$ is independent of $\boldsymbol{x}$ and is always positive semi-definite, $f_k$ is convex and its gradient is Lipschitz continuous. A valid Lipschitz constant for each client $k$ is

$$L_k = \frac{2}{n} \left\|\nabla^2 f_k(\boldsymbol{x})\right\|_2 = \frac{2}{n} \left\|\boldsymbol{H}_k\right\|_2^2.$$

We can then choose $L = \max_k \{L_k\}$. $\qquad\square$

**Property 2.** *If Assumption 1 holds, then the global objective $P : \mathbb{R}^d \to \mathbb{R} \cup \{+\infty\}$ is bounded below; that is, there exists $P^\star$ with $P(\boldsymbol{x}) \geqslant P^\star > -\infty$ for all $\boldsymbol{x} \in \mathbb{R}^d$.*

*Proof.* By Assumption 1, $g(\boldsymbol{x})$ is bounded below, and we denote the bound by $g^\star$. Since $f_k(\boldsymbol{x}) \geqslant 0$ for all $k$,

$$P(\boldsymbol{x}) = \sum_{k=1}^n f_k(\boldsymbol{x}) + g(\boldsymbol{x}) \geqslant g(\boldsymbol{x}) \geqslant g^\star > -\infty.$$

Hence $P(\boldsymbol{x})$ is bounded below. $\qquad\square$

## E.2  KEY LEMMAS

This section presents two standard lemmas that are used in the subsequent analysis. We include their proofs for completeness.

**Lemma 1** (Three-point identity). *For any vectors $\boldsymbol{a}, \boldsymbol{b}, \boldsymbol{c} \in \mathbb{R}^d$, the following identity holds:*

$$\|\boldsymbol{a} - \boldsymbol{c}\|_2^2 - \|\boldsymbol{b} - \boldsymbol{c}\|_2^2 = 2\langle \boldsymbol{a} - \boldsymbol{b}, \boldsymbol{a} - \boldsymbol{c}\rangle - \|\boldsymbol{a} - \boldsymbol{b}\|_2^2.$$

*Proof.* We prove the identity by starting from the right-hand side and rearranging the terms:

$$
\begin{aligned}
2\langle \boldsymbol{a} - \boldsymbol{b}, \boldsymbol{a} - \boldsymbol{c}\rangle - \|\boldsymbol{a} - \boldsymbol{b}\|_2^2 &= 2\langle \boldsymbol{a} - \boldsymbol{b}, \boldsymbol{a} - \boldsymbol{c}\rangle - \langle \boldsymbol{a} - \boldsymbol{b}, \boldsymbol{a} - \boldsymbol{b}\rangle \\
&= \langle \boldsymbol{a} - \boldsymbol{b}, 2(\boldsymbol{a} - \boldsymbol{c}) - (\boldsymbol{a} - \boldsymbol{b})\rangle \\
&= \langle \boldsymbol{a} - \boldsymbol{b}, \boldsymbol{a} + \boldsymbol{b} - 2\boldsymbol{c}\rangle \\
&= \langle (\boldsymbol{a} - \boldsymbol{c}) - (\boldsymbol{b} - \boldsymbol{c}), (\boldsymbol{a} - \boldsymbol{c}) + (\boldsymbol{b} - \boldsymbol{c})\rangle \\
&= \|\boldsymbol{a} - \boldsymbol{c}\|_2^2 - \|\boldsymbol{b} - \boldsymbol{c}\|_2^2.
\end{aligned}
$$

The last equality follows from the difference of squares identity. $\square$

**Lemma 2** (Descent lemma). *Let the function $f : \mathbb{R}^d \to \mathbb{R}$ be $L$-smooth. Then for any $\boldsymbol{x}_1, \boldsymbol{x}_2 \in \mathbb{R}^d$, we have*

$$
f(\boldsymbol{x}_1) - f(\boldsymbol{x}_2) - \langle \nabla f(\boldsymbol{x}_2), \boldsymbol{x}_1 - \boldsymbol{x}_2\rangle \leqslant \frac{L}{2}\|\boldsymbol{x}_1 - \boldsymbol{x}_2\|_2^2.
$$

*Proof.* By the fundamental theorem of calculus,

$$
f(\boldsymbol{x}_1) - f(\boldsymbol{x}_2) = \int_0^1 \langle \nabla f(\boldsymbol{x}_2 + t(\boldsymbol{x}_1 - \boldsymbol{x}_2)), \boldsymbol{x}_1 - \boldsymbol{x}_2\rangle \, \mathrm{d}t.
$$

Therefore,

$$
\begin{aligned}
&f(\boldsymbol{x}_1) - f(\boldsymbol{x}_2) - \langle \nabla f(\boldsymbol{x}_2), \boldsymbol{x}_1 - \boldsymbol{x}_2\rangle \\
&= \int_0^1 \langle \nabla f(\boldsymbol{x}_2 + t(\boldsymbol{x}_1 - \boldsymbol{x}_2)) - \nabla f(\boldsymbol{x}_2), \boldsymbol{x}_1 - \boldsymbol{x}_2\rangle \, \mathrm{d}t.
\end{aligned}
$$

Applying the Cauchy-Schwarz inequality, we have

$$
\begin{aligned}
&\langle \nabla f(\boldsymbol{x}_2 + t(\boldsymbol{x}_1 - \boldsymbol{x}_2)) - \nabla f(\boldsymbol{x}_2), \boldsymbol{x}_1 - \boldsymbol{x}_2\rangle \\
&\leqslant \|\nabla f(\boldsymbol{x}_2 + t(\boldsymbol{x}_1 - \boldsymbol{x}_2)) - \nabla f(\boldsymbol{x}_2)\|_2 \cdot \|\boldsymbol{x}_1 - \boldsymbol{x}_2\|_2.
\end{aligned}
$$

Using the $L$-smoothness of $f$, we can bound the first term:

$$
\|\nabla f(\boldsymbol{x}_2 + t(\boldsymbol{x}_1 - \boldsymbol{x}_2)) - \nabla f(\boldsymbol{x}_2)\|_2 \leqslant L\|(\boldsymbol{x}_2 + t(\boldsymbol{x}_1 - \boldsymbol{x}_2)) - \boldsymbol{x}_2\|_2 = Lt\|\boldsymbol{x}_1 - \boldsymbol{x}_2\|_2.
$$

Substituting this back into the integral, we get

$$
f(\boldsymbol{x}_1) - f(\boldsymbol{x}_2) - \langle \nabla f(\boldsymbol{x}_2), \boldsymbol{x}_1 - \boldsymbol{x}_2\rangle \leqslant \int_0^1 Lt\|\boldsymbol{x}_1 - \boldsymbol{x}_2\|_2^2 \, \mathrm{d}t = \frac{L}{2}\|\boldsymbol{x}_1 - \boldsymbol{x}_2\|_2^2.
$$

This completes the proof. $\square$

### E.3 MAIN PROOF

Using the notation from (11), the optimization problem (3) can be equivalently written as

$$
\begin{aligned}
\min_{\boldsymbol{x}, \{\boldsymbol{x}_k\}} \quad & \sum_{k=1}^n f_k(\boldsymbol{x}_k) + g(\boldsymbol{x}) \\
\text{s.t.} \quad & \boldsymbol{x}_k - \boldsymbol{x} = \boldsymbol{0}, \quad k = 1, \dots, n.
\end{aligned}
\tag{12}
$$

Its augmented Lagrangian is

$$
\mathcal{L}_\rho(\boldsymbol{x}, \{\boldsymbol{x}_k\}, \{\boldsymbol{\lambda}_k\}) = \sum_{k=1}^n L_k(\boldsymbol{x}, \boldsymbol{x}_k, \boldsymbol{\lambda}_k) + g(\boldsymbol{x}),
\tag{13}
$$

$$
\text{where} \quad L_k(\boldsymbol{x}, \boldsymbol{x}_k, \boldsymbol{\lambda}_k) = f_k(\boldsymbol{x}_k) + \langle \boldsymbol{\lambda}_k, \boldsymbol{x}_k - \boldsymbol{x}\rangle + \frac{\rho}{2}\|\boldsymbol{x}_k - \boldsymbol{x}\|_2^2.
$$

The ADMM updates of `FedDeblur` algorithm can be expressed as

$$
\boldsymbol{x}^{t+1} = \arg\min_{\boldsymbol{x}} \left\{ g(\boldsymbol{x}) + \sum_{k=1}^n \left( \langle \boldsymbol{\lambda}_k^t, \boldsymbol{x}_k^t - \boldsymbol{x}\rangle + \frac{\rho}{2}\|\boldsymbol{x}_k^t - \boldsymbol{x}\|_2^2 \right) \right\};
\tag{14}
$$

$$\boldsymbol{x}_k^{t+1} = \arg\min_{\boldsymbol{x}_k} L_k(\boldsymbol{x}^{t+1}, \boldsymbol{x}_k, \boldsymbol{\lambda}_k^t) \quad \text{for } k = 1, \ldots, n \text{ in parallel;} \tag{15}$$

$$\boldsymbol{\lambda}_k^{t+1} = \boldsymbol{\lambda}_k^t + \rho(\boldsymbol{x}_k^{t+1} - \boldsymbol{x}^{t+1}) \quad \text{for } k = 1, \ldots, n \text{ in parallel.} \tag{16}$$

The first-order optimality conditions for the subproblems (14) and (15) are, respectively,

$$\boldsymbol{0} \in \partial g\left(\boldsymbol{x}^{t+1}\right) - \sum_{k=1}^n \left(\boldsymbol{\lambda}_k^t + \rho\left(\boldsymbol{x}^{t+1} - \boldsymbol{x}_k^t\right)\right),$$

$$\nabla f_k\left(\boldsymbol{x}_k^{t+1}\right) + \boldsymbol{\lambda}_k^t + \rho\left(\boldsymbol{x}_k^{t+1} - \boldsymbol{x}^{t+1}\right) = \boldsymbol{0}, \quad k = 1, \ldots, n.$$

By substituting the expression for $\boldsymbol{\lambda}_k$ from (16), we get

$$\sum_{k=1}^n \left(\boldsymbol{\lambda}_k^{t+1} - \rho(\boldsymbol{x}_k^{t+1} - \boldsymbol{x}_k^t)\right) \in \partial g(\boldsymbol{x}^{t+1}), \tag{17}$$

$$\boldsymbol{\lambda}_k^{t+1} = -\nabla f_k\left(\boldsymbol{x}_k^{t+1}\right), \quad k = 1, \ldots, n. \tag{18}$$

**Lemma 3** (Sufficient descent)**.** *If Assumption 1 holds, then for any $t \geqslant 0$, the iterates satisfy*

$$\mathcal{L}_\rho^{t+1} - \mathcal{L}_\rho^t \leqslant -\frac{n\rho}{2}\left\|\boldsymbol{x}^{t+1} - \boldsymbol{x}^t\right\|_2^2 - \sum_{k=1}^n \left(\frac{\rho - L}{2} - \frac{L^2}{\rho}\right)\left\|\boldsymbol{x}_k^{t+1} - \boldsymbol{x}_k^t\right\|_2^2,$$

*where $\mathcal{L}_\rho^t = \mathcal{L}_\rho(\boldsymbol{x}^t, \{\boldsymbol{x}_k^t\}, \{\boldsymbol{\lambda}_k^t\})$.*

*Proof.* We decompose the difference $\mathcal{L}_\rho^{t+1} - \mathcal{L}_\rho^t$ into three parts:

$$\mathcal{L}_\rho^{t+1} - \mathcal{L}_\rho^t = p_1^t + p_2^t + p_3^t,$$

where

$$p_1^t := \mathcal{L}_\rho\left(\boldsymbol{x}^{t+1}, \{\boldsymbol{x}_k^t\}, \{\boldsymbol{\lambda}_k^t\}\right) - \mathcal{L}_\rho\left(\boldsymbol{x}^t, \{\boldsymbol{x}_k^t\}, \{\boldsymbol{\lambda}_k^t\}\right),$$

$$p_2^t := \mathcal{L}_\rho\left(\boldsymbol{x}^{t+1}, \{\boldsymbol{x}_k^{t+1}\}, \{\boldsymbol{\lambda}_k^t\}\right) - \mathcal{L}_\rho\left(\boldsymbol{x}^{t+1}, \{\boldsymbol{x}_k^t\}, \{\boldsymbol{\lambda}_k^t\}\right),$$

$$p_3^t := \mathcal{L}_\rho\left(\boldsymbol{x}^{t+1}, \{\boldsymbol{x}_k^{t+1}\}, \{\boldsymbol{\lambda}_k^{t+1}\}\right) - \mathcal{L}_\rho\left(\boldsymbol{x}^{t+1}, \{\boldsymbol{x}_k^{t+1}\}, \{\boldsymbol{\lambda}_k^t\}\right).$$

**Bounding $p_1^t$.** Let $G(\boldsymbol{x})$ be the objective for the $\boldsymbol{x}$-update. We can write it as

$$G(\boldsymbol{x}) = g(\boldsymbol{x}) - \sum_{k=1}^n \left\langle \boldsymbol{\lambda}_k^t + \rho\boldsymbol{x}_k^t, \boldsymbol{x} \right\rangle + \frac{n\rho}{2}\|\boldsymbol{x}\|_2^2 + \text{const}.$$

Since $g$ is convex, $G(\boldsymbol{x})$ is $n\rho$-strongly convex. By the optimality of the update (14), we have

$$G\left(\boldsymbol{x}^{t+1}\right) \leqslant G\left(\boldsymbol{x}^t\right) - \frac{n\rho}{2}\left\|\boldsymbol{x}^{t+1} - \boldsymbol{x}^t\right\|_2^2.$$

Therefore,

$$p_1^t = G\left(\boldsymbol{x}^{t+1}\right) - G\left(\boldsymbol{x}^t\right) \leqslant -\frac{n\rho}{2}\left\|\boldsymbol{x}^{t+1} - \boldsymbol{x}^t\right\|_2^2.$$

**Bounding $p_2^t$.** We have $p_2^t = \sum_{k=1}^n Q_k^t$, where

$$Q_k^t = f_k\left(\boldsymbol{x}_k^{t+1}\right) - f_k\left(\boldsymbol{x}_k^t\right) + \left\langle \boldsymbol{\lambda}_k^t, \boldsymbol{x}_k^{t+1} - \boldsymbol{x}_k^t \right\rangle + \frac{\rho}{2}\left(\left\|\boldsymbol{x}_k^{t+1} - \boldsymbol{x}^{t+1}\right\|_2^2 - \left\|\boldsymbol{x}_k^t - \boldsymbol{x}^{t+1}\right\|_2^2\right)$$

$$= f_k\left(\boldsymbol{x}_k^{t+1}\right) - f_k\left(\boldsymbol{x}_k^t\right) + \left\langle \boldsymbol{\lambda}_k^t, \boldsymbol{x}_k^{t+1} - \boldsymbol{x}_k^t \right\rangle$$

$$\quad + \rho\left\langle \boldsymbol{x}_k^{t+1} - \boldsymbol{x}_k^t, \boldsymbol{x}_k^{t+1} - \boldsymbol{x}^{t+1} \right\rangle - \frac{\rho}{2}\left\|\boldsymbol{x}_k^{t+1} - \boldsymbol{x}_k^t\right\|_2^2 \quad \text{(by Lemma 1)}$$

$$= f_k\left(\boldsymbol{x}_k^{t+1}\right) - f_k\left(\boldsymbol{x}_k^t\right) + \left\langle \boldsymbol{\lambda}_k^{t+1}, \boldsymbol{x}_k^{t+1} - \boldsymbol{x}_k^t \right\rangle - \frac{\rho}{2}\left\|\boldsymbol{x}_k^{t+1} - \boldsymbol{x}_k^t\right\|_2^2 \quad \text{(by (16))}$$

$$= f_k\left(\boldsymbol{x}_k^{t+1}\right) - f_k\left(\boldsymbol{x}_k^t\right) - \left\langle \nabla f_k\left(\boldsymbol{x}_k^{t+1}\right), \boldsymbol{x}_k^{t+1} - \boldsymbol{x}_k^t \right\rangle - \frac{\rho}{2}\left\|\boldsymbol{x}_k^{t+1} - \boldsymbol{x}_k^t\right\|_2^2 \quad \text{(by (18))}$$

$$\leqslant \frac{L - \rho}{2}\left\|\boldsymbol{x}_k^{t+1} - \boldsymbol{x}_k^t\right\|_2^2 \quad \text{(by Lemma 2).}$$

Thus,

$$p_2^t \leqslant -\frac{\rho - L}{2} \sum_{k=1}^{n} \left\| \boldsymbol{x}_k^{t+1} - \boldsymbol{x}_k^t \right\|_2^2.$$

**Bounding $p_3^t$.** From (16), we have $\boldsymbol{\lambda}_k^{t+1} - \boldsymbol{\lambda}_k^t = \rho \left( \boldsymbol{x}_k^{t+1} - \boldsymbol{x}^{t+1} \right)$. Using this and Property 1:

$$p_3^t = \sum_{k=1}^{n} \left\langle \boldsymbol{\lambda}_k^{t+1} - \boldsymbol{\lambda}_k^t, \boldsymbol{x}_k^{t+1} - \boldsymbol{x}^{t+1} \right\rangle = \frac{1}{\rho} \sum_{k=1}^{n} \left\| \boldsymbol{\lambda}_k^{t+1} - \boldsymbol{\lambda}_k^t \right\|_2^2$$

$$= \frac{1}{\rho} \sum_{k=1}^{n} \left\| \nabla f_k \left( \boldsymbol{x}_k^{t+1} \right) - \nabla f_k \left( \boldsymbol{x}_k^t \right) \right\|_2^2 \leqslant \frac{L^2}{\rho} \sum_{k=1}^{n} \left\| \boldsymbol{x}_k^{t+1} - \boldsymbol{x}_k^t \right\|_2^2.$$

Combining these bounds yields the result:

$$\mathcal{L}_\rho^{t+1} - \mathcal{L}_\rho^t = p_1^t + p_2^t + p_3^t \leqslant -\frac{n\rho}{2} \left\| \boldsymbol{x}^{t+1} - \boldsymbol{x}^t \right\|_2^2 - \sum_{k=1}^{n} \left( \frac{\rho - L}{2} - \frac{L^2}{\rho} \right) \left\| \boldsymbol{x}_k^{t+1} - \boldsymbol{x}_k^t \right\|_2^2.$$

This completes the proof. $\qquad\square$

**Lemma 4** (Asymptotic regularity). *If Assumption 1 holds and $\rho > 2L$, the sequence of augmented Lagrangians $\{\mathcal{L}_\rho^t\}_{t=0}^\infty$ converges, and*

$$\lim_{t\to\infty} \left\| \boldsymbol{x}^{t+1} - \boldsymbol{x}^t \right\|_2 = \lim_{t\to\infty} \left\| \boldsymbol{x}_k^{t+1} - \boldsymbol{x}_k^t \right\|_2 = \lim_{t\to\infty} \left\| \boldsymbol{\lambda}_k^{t+1} - \boldsymbol{\lambda}_k^t \right\|_2 = \lim_{t\to\infty} \left\| \boldsymbol{x}_k^t - \boldsymbol{x}^t \right\|_2 = 0$$

*for each $k = 1, \ldots, n$.*

*Proof.* First, we show that $\{\mathcal{L}_\rho^t\}$ is bounded below. By Lemma 2,

$$f_k \left( \boldsymbol{x}^t \right) - f_k \left( \boldsymbol{x}_k^t \right) \leqslant \left\langle \nabla f_k \left( \boldsymbol{x}_k^t \right), \boldsymbol{x}^t - \boldsymbol{x}_k^t \right\rangle + \frac{L}{2} \left\| \boldsymbol{x}^t - \boldsymbol{x}_k^t \right\|_2^2.$$

Using (18), this becomes

$$f_k \left( \boldsymbol{x}^t \right) - f_k \left( \boldsymbol{x}_k^t \right) \leqslant \left\langle -\boldsymbol{\lambda}_k^t, \boldsymbol{x}^t - \boldsymbol{x}_k^t \right\rangle + \frac{L}{2} \left\| \boldsymbol{x}^t - \boldsymbol{x}_k^t \right\|_2^2.$$

Thus, for $\rho > L$,

$$\mathcal{L}_\rho^t = g \left( \boldsymbol{x}^t \right) + \sum_{k=1}^{n} \left( f_k \left( \boldsymbol{x}_k^t \right) + \left\langle \boldsymbol{\lambda}_k^t, \boldsymbol{x}_k^t - \boldsymbol{x}^t \right\rangle + \frac{\rho}{2} \left\| \boldsymbol{x}_k^t - \boldsymbol{x}^t \right\|_2^2 \right)$$

$$\geqslant g \left( \boldsymbol{x}^t \right) + \sum_{k=1}^{n} f_k \left( \boldsymbol{x}^t \right) \geqslant P^\star > -\infty \quad \text{(by Property 2)}.$$

This shows that the sequence $\{\mathcal{L}_\rho^t\}$ is bounded below. From Lemma 3, if $\rho$ is large enough such that $\frac{\rho - L}{2} - \frac{L^2}{\rho} > 0$ (which holds when $\rho > 2L$), then $\{\mathcal{L}_\rho^t\}$ is monotonically decreasing. So it converges.

The convergence of $\{\mathcal{L}_\rho^t\}$ implies $\lim_{t\to\infty} \left( \mathcal{L}_\rho^t - \mathcal{L}_\rho^{t+1} \right) = 0$. From the inequality in Lemma 3, this requires that both non-negative terms on the right-hand side go to zero:

$$\lim_{t\to\infty} \left\| \boldsymbol{x}^{t+1} - \boldsymbol{x}^t \right\|_2 = 0 \quad \text{and} \quad \lim_{t\to\infty} \left\| \boldsymbol{x}_k^{t+1} - \boldsymbol{x}_k^t \right\|_2 = 0, \quad \forall k.$$

The $L$-smoothness of $f_k$ implies

$$\left\| \boldsymbol{\lambda}_k^{t+1} - \boldsymbol{\lambda}_k^t \right\|_2 = \left\| \nabla f_k \left( \boldsymbol{x}_k^{t+1} \right) - \nabla f_k \left( \boldsymbol{x}_k^t \right) \right\|_2 \leqslant L \left\| \boldsymbol{x}_k^{t+1} - \boldsymbol{x}_k^t \right\|_2,$$

so

$$\lim_{t\to\infty} \left\| \boldsymbol{\lambda}_k^{t+1} - \boldsymbol{\lambda}_k^t \right\|_2 = 0.$$

Finally, since

$$\boldsymbol{x}_k^{t+1} - \boldsymbol{x}^{t+1} = \frac{1}{\rho} (\boldsymbol{\lambda}_k^{t+1} - \boldsymbol{\lambda}_k^t),$$

the consensus residual also vanishes:

$$\lim_{t \to \infty} \left\| \boldsymbol{x}_k^t - \boldsymbol{x}^t \right\|_2 = 0.$$

This completes the proof. $\qquad \square$

**Theorem 2.** *Let* $\left\{ \left( \boldsymbol{x}^t, \{\boldsymbol{x}_k^t\}, \{\boldsymbol{\lambda}_k^t\} \right) \right\}$ *be the sequence generated by iterations (14-16). If Assumption 1 holds and* $\rho > 2L$*, then any accumulation point* $(\boldsymbol{x}^\infty, \{\boldsymbol{x}_k^\infty\}, \{\boldsymbol{\lambda}_k^\infty\})$ *of this sequence is a Karush-Kuhn-Tucker (KKT) point of the optimization problem* (12)*, satisfying*

$$\boldsymbol{x}_k^\infty = \boldsymbol{x}^\infty, \quad k = 1, \ldots, n;$$
$$\nabla f_k (\boldsymbol{x}_k^\infty) + \boldsymbol{\lambda}_k^\infty = \boldsymbol{0}, \quad k = 1, \ldots, n;$$
$$\boldsymbol{0} \in \partial g (\boldsymbol{x}^\infty) - \sum_{k=1}^n \boldsymbol{\lambda}_k^\infty.$$

*Proof.* Let $(\boldsymbol{x}^\infty, \{\boldsymbol{x}_k^\infty\}, \{\boldsymbol{\lambda}_k^\infty\})$ be an accumulation point of the sequence. Then there exists a subsequence, indexed by $\{t_s\}_{s=1}^\infty$, such that

$$\lim_{s \to \infty} \left( \boldsymbol{x}^{t_s}, \{\boldsymbol{x}_k^{t_s}\}, \{\boldsymbol{\lambda}_k^{t_s}\} \right) = (\boldsymbol{x}^\infty, \{\boldsymbol{x}_k^\infty\}, \{\boldsymbol{\lambda}_k^\infty\}).$$

From Lemma 4, the consensus residual converges to zero, $\lim_{t \to \infty} (\boldsymbol{x}_k^t - \boldsymbol{x}^t) = \boldsymbol{0}$. Taking the limit along the subsequence gives

$$\boldsymbol{x}_k^\infty - \boldsymbol{x}^\infty = \boldsymbol{0}, \quad \forall k.$$

The optimality condition for the $\boldsymbol{x}_k$-update (18) gives $\nabla f_k \left( \boldsymbol{x}_k^{t_s+1} \right) + \boldsymbol{\lambda}_k^{t_s+1} = \boldsymbol{0}$. As $f_k$ is $L$-smooth, its gradient $\nabla f_k$ is continuous. Taking the limit as $s \to \infty$ and using the results of Lemma 4, $\boldsymbol{x}_k^{t_s} \to \boldsymbol{x}_k^\infty, \boldsymbol{\lambda}_k^{t_s} \to \boldsymbol{\lambda}_k^\infty$, then

$$\nabla f_k (\boldsymbol{x}_k^\infty) + \boldsymbol{\lambda}_k^\infty = \boldsymbol{0}, \quad \forall k.$$

Finally, from the optimality condition for the $\boldsymbol{x}$-update (17), we have

$$\sum_{k=1}^n \left( \boldsymbol{\lambda}_k^{t_s+1} - \rho \left( \boldsymbol{x}_k^{t_s+1} - \boldsymbol{x}_k^{t_s} \right) \right) \in \partial g \left( \boldsymbol{x}^{t_s+1} \right).$$

As $s \to \infty$, the left-hand side converges to $\sum_{k=1}^n \boldsymbol{\lambda}_k^\infty$ due to Lemma 4. Since $g$ is proper, convex, and lower semi-continuous, its subdifferential operator $\partial g$ has a closed graph. Therefore, we can take the limit inside the subdifferential, which yields

$$\sum_{k=1}^n \boldsymbol{\lambda}_k^\infty \in \partial g (\boldsymbol{x}^\infty).$$

This confirms that the accumulation point is a KKT point of the original problem. $\qquad \square$

STATEMENT ON THE USE OF LARGE LANGUAGE MODELS

In adherence to the conference guidelines, we clarify the role of Large Language Models (LLMs) in the preparation of this manuscript. We utilized an LLM as a general-purpose assistive tool, primarily for language refinement and grammatical correction. The core scientific contributions of this work—including the initial research ideation, the formulation of the federated multi-observation deblurring problem, the development of the `FedDeblur` framework, all theoretical analyses, the design and execution of experiments, and the interpretation of results—were conceived and carried out exclusively by the human authors. The LLM was employed to improve the clarity and readability of the text. The authors take full responsibility for all content presented in this paper, including its scientific accuracy and integrity. The LLM is not considered an author of this work.

