# OpenReview forum: "Federated Deblurring as a Jigsaw Puzzle: A Privacy-Preserving Consensus Framework"
_ICLR.cc/2026/Conference — Submitted to ICLR 2026_

### Official Review · Reviewer_eUZt · 2025-10-28

**Soundness:** 2
**Presentation:** 3
**Contribution:** 3
**Rating:** 4
**Confidence:** 1

**Summary:**

The paper studies federated deblurring with multiple observations. Several clients each hold a blurred view of the same scene, and some views cover only part of the scene. The goal is to recover a clean global image without sending raw images, kernels, or masks. The proposed framework, FedDeblur, follows a consensus design: each client performs the data fidelity step locally, and the server applies a light proximal update to a shared proxy that protects privacy. The method supports partial field of view, allows plug and play priors on the server, and works with partial client participation. Under convex regularizers, the authors also provide convergence guarantees.

**Strengths:**

The paper targets a relevant privacy-constrained collaboration setting and defines it clearly. The consensus design is simple and practical: data-fidelity updates stay on device, priors are applied on the server, and only a sanitized proxy is communicated, which creates a natural privacy boundary. The framework is modular and supports both classical TV priors and plug-and-play denoisers without changing the interface. It also handles partial field of view and partial client participation with the same protocol. The algorithm admits closed-form local updates and a lightweight server step, and the authors provide convergence guarantees under convex regularizers. Empirical results are close to a centralized oracle and stronger than non-collaborative and Fed-averaging baselines, with useful robustness studies over participation rate, noise level, and hyperparameters.

**Weaknesses:**

I acknowledge that I am not a domain expert in federated imaging, but from a generalist technical perspective several aspects feel underdeveloped. The practical value proposition and concrete deployment scenarios are not specified clearly enough to justify the added complexity of the federated setup; the privacy claim remains qualitative, lacking a precise threat model, leakage experiments, or formal guarantees. Basic systems evidence is missing, including latency per image, communication volume per round, and memory or compute footprint, which makes operational feasibility hard to judge. The experiments appear toy-scale, relying on low-resolution synthetic data, regular grid crops, and known blur kernels, with no tests on irregular partial views, blind deblurring, or diverse real devices. Robustness to stragglers, unreliable networks, or adversarial and low-quality clients is not examined. Clearer reproducibility materials and a brief end-to-end walkthrough would also help non-specialists and practitioners validate and adopt the method.

**Questions:**

I’m not a domain expert here, so I’d appreciate clarification on a few essentials: the concrete deployment scenarios where the federated design clearly beats a centralized upload; a precise threat model plus any attack tests or formal guarantees showing the shared proxy doesn’t leak content or device fingerprints; basic efficiency numbers (per-round communication, latency, memory/compute, time-to-convergence); and broader validation on real multi-device data, blind deblurring, irregular/sparse FOV, and higher resolutions, including robustness to stragglers or adversarial/low-quality clients and the stability of the PnP variant. I’ll weigh your rebuttal and the other reviewers’ input before finalizing my score.

---

> ### Author Response · Authors · 2025-12-04
>
> We thank the reviewer for the insightful comments, particularly regarding systems-level evidence and experimental realism. We have addressed these points thoroughly in the revised manuscript.
>
> **1. Systems Evidence: Efficiency and Cost**
>
> To provide concrete evidence of operational feasibility, we added **Appendix D.6** with a detailed efficiency analysis. We compared the per-round computation time and total convergence time of `FedDeblur` against the baseline `FedAvgDeblur`. As shown in **Table A** below, our method is significantly more efficient. Specifically, on the "Barbara" color image ($n=10$), `FedDeblur` reaches a high quality of 37dB in just **0.37 seconds** (14 rounds), whereas `FedAvgDeblur` takes **4.35 seconds** (82 rounds). This represents a speedup of over **10$\times$** in total time to convergence, demonstrating the practical value of our consensus optimization.
>
> **Table A: Runtime and convergence speed comparison ($n=10$).** The *Avg. Time per Round* is computed over the entire dataset. The *Rounds / Total Time* to reach targets are measured on the color "Barbara" image.
>
> | Method        | Avg. Time per Round (s) |            | Rounds / Total Time (s) to reach target |               |               |               |               |                |
> | :------------ | :---------------------: | :--------: | :-------------------------------------: | :-----------: | :-----------: | :-----------: | :-----------: | :------------: |
> |               |        **Gray**         | **Color**  |              **PSNR 35dB**              | **PSNR 37dB** | **PSNR 38dB** | **SSIM 0.95** | **SSIM 0.97** | **SSIM 0.975** |
> | **FedDeblur** |       **0.0129**        | **0.0275** |              **12 / 0.32**              | **14 / 0.37** | **20 / 0.53** | **12 / 0.32** | **15 / 0.40** | **42 / 1.11**  |
> | FedAvgDeblur  |         0.0370          |   0.0560   |                64 / 3.41                |   82 / 4.35   |       -       |   65 / 3.46   |   84 / 4.45   |       -        |
>
> **2. Experimental Scale and Realism**
>
> We have expanded our evaluation beyond "toy-scale" datasets to include high-resolution and real-world scenarios in **Appendix D.7**:
>
> *   **High-Resolution Data:** We tested on high-definition images from the **DIV2K dataset** ($2040 \times 1356$ pixels). We simulated a scenario with $n=16$ clients arranged in a $4 \times 4$ grid, applying large $25 \times 25$ synthetic blur kernels. The results in **Table B** show that `FedDeblur` maintains performance very close to the centralized oracle, proving its scalability to large images and complex degradations.
> *   **Real-World Data:** We utilized the **DTU MVS dataset**, using 5 real images captured from adjacent viewpoints to represent 5 clients. These images naturally contain variations in lighting and perspective. Using basic ECC alignment, `FedDeblur` successfully integrated these views into a cohesive, high-quality image (**Figure 16**), confirming robustness to real-world data heterogeneity.
>
> **Table B: Quantitative comparison on high-resolution images from the DIV2K dataset ($n=16$).**
>
> | Method        | Image ID |  PSNR (dB)  |     SSIM     |
> | :------------ | :------: | :---------: | :----------: |
> | **FedDeblur** |   0808   |   30.1567   |   0.890674   |
> |               |   0865   |   30.3899   | **0.874092** |
> | **CenDeblur** |   0808   | **30.7273** | **0.893924** |
> | (Oracle)      |   0865   | **30.5367** |   0.869411   |
>
> **3. Robustness to Stragglers**
>
> We have explicitly addressed robustness to stragglers and unreliable networks in **Appendix D.3**. We simulated partial client participation, where only a subset of clients participate in each round. The results demonstrate that `FedDeblur` is highly robust; even if only 2 out of 10 clients are active per round, the algorithm successfully converges to a high-quality solution, albeit with slightly slower convergence than full participation.

---

### Official Review · Reviewer_2hPc · 2025-10-29

**Soundness:** 3
**Presentation:** 3
**Contribution:** 2
**Rating:** 2
**Confidence:** 4

**Summary:**

This paper addresses the problem of recovering a sharp picture from multiple crops of that image (each one blurred by its own blur kernel), possibly captured by multiple devices, in a federated computation framework, addressing in particular privacy issues.

**Strengths:**

The paper addresses what is, as far as I know, a new problem, and the proposed approach appears to be sound, with classical components such as ADMM, proximal updates, etc., as well as a genuine effort to propose a privacy preserving approach in a distributed effort, and some theoretical analysis in the case where all clients have access to (a blurry version of) the whole image.

**Weaknesses:**

It is difficult for me to envision a practical application scenario for the proposed approach. Unless I missed something, the blur kernels are assumed to be known. This may be true for astronomy or microscopy applications, but they typically are unknown for the photography applications that appear to be the focus of the paper. In addition, defocus blur normally varies within an image, including image crops unless they are quite small.  Motion blur often does, and it is unclear how the multiples images used as input are supposed to be synchronized in a dynamic setting. Multiple devices (or the same device over time) induce images motions that are not accounted for in the paper, where, unless I am mistaken, all input images are assumed to be prealigned. Of course, one could just distribute deblurring of a single image with (known) spatially varying levels of blur across multiple processors processing individual crops, with one kernel associated with each crop, but the images under consideration are very small (256x256) and it is unclear why a federated algorithm is needed for such images. It should also be noted that the CenDeblur baseline appears to do at least as well as the proposed method in quantitative evaluations. Finally, there is no qualitative evaluation on real images.

**Questions:**

I would appreciate if the authors clarified the issues raised in the "weaknesses" section of this review.

---

> ### Author Response · Authors · 2025-12-04
>
> We thank the reviewer for the thoughtful feedback. We appreciate your recognition of the novelty of the problem and the soundness of the proposed approach. We address your specific concerns regarding practical application below.
>
> **1. Non-blind Assumption**
>
> We acknowledge that assuming known blur kernels is a simplification. However, non-blind deblurring remains a significant and active research area [1-3], often serving as the core solver within alternating minimization schemes for blind deblurring. Our work establishes the fundamental optimization framework for the federated setting. Extending this to handle blind deblurring scenarios is a planned and logical next step for future research.
>
> **2. Alignment and Synchronization**
>
> We assume that input images are pre-aligned, which is a standard prerequisite in multi-image processing pipelines. In practical scenarios, geometric and photometric differences can be corrected using mature registration algorithms [4-9] or calibration targets.
>
> To demonstrate that our method is robust to real-world alignment issues, we added a new experiment in **Appendix D.7**. We used the **DTU MVS dataset**, selecting 5 real images of a scene captured from adjacent viewpoints with naturally varying perspective and lighting. By applying standard, lightweight pre-processing—specifically the Enhanced Correlation Coefficient (ECC) algorithm for spatial alignment and histogram matching for color correction—`FedDeblur` successfully fused these views into a high-quality global image (see **Figure 16**). This confirms that standard pre-processing is sufficient for our algorithm to operate effectively on real-world data.
>
> **3. Justification for Federated Approach and Image Scale**
>
> The primary motivation for a federated approach is **privacy**, which is independent of image size. If the image data contains sensitive information (e.g., facial recognition data, medical imaging, or surveillance of private property), transmitting raw pixels to a central server is often prohibited by regulations, regardless of whether the image is $256 \times 256$ or $4K$. Our framework allows for collaborative restoration without raw data ever leaving the local device.
>
> Furthermore, to address the concern that we only tested on small images, we have extended our evaluation to high-resolution data in **Appendix D.7**. We tested on **DIV2K** images ($2040 \times 1356$) using a simulated system of $n=16$ clients with large $25 \times 25$ blur kernels. As shown in **Table A**, `FedDeblur` performs exceptionally well at this scale, achieving results comparable to the centralized oracle.
>
> **Table A: Quantitative comparison on high-resolution images from the DIV2K dataset ($n=16$).**
>
> | Method        | Image ID |  PSNR (dB)  |     SSIM     |
> | :------------ | :------: | :---------: | :----------: |
> | **FedDeblur** |   0808   |   30.1567   |   0.890674   |
> |               |   0865   |   30.3899   | **0.874092** |
> | **CenDeblur** |   0808   | **30.7273** | **0.893924** |
> | (Oracle)      |   0865   | **30.5367** |   0.869411   |
>
> **4. Performance Comparison with CenDeblur**
>
> We wish to clarify that `CenDeblur` is designed as the **centralized oracle** (an idealized upper bound). It assumes full access to all raw data and masks without any privacy constraints. The fact that `FedDeblur` achieves performance very close to `CenDeblur` is a significant **strength** of our method. It demonstrates that our federated consensus mechanism successfully solves the "jigsaw puzzle" and recovers the image almost as well as if all data were centrally available, but does so while strictly preserving data privacy.

---

> > ### Author Response · Authors · 2025-12-04
> >
> > **References:**
> >
> > 1.  Dong, J., Roth, S., & Schiele, B. (2021). Dwdn: Deep wiener deconvolution network for non-blind image deblurring. *IEEE Transactions on Pattern Analysis and Machine Intelligence*, 44(12), 9960-9976.
> > 2.  Chen, L., Zhang, J., Li, Z., Wei, Y., Fang, F., Ren, J., & Pan, J. (2024). Deep Richardson–Lucy deconvolution for low-light image deblurring. *International Journal of Computer Vision*, 132(2), 428-445.
> > 3.  Li, J., & Wang, C. (2025). Efficient diffusion posterior sampling for noisy inverse problems. *SIAM Journal on Imaging Sciences*, 18(2), 1468-1492.
> > 4.  Wei, G., Lan, C., Zeng, W., Zhang, Z., & Chen, Z. (2021). Toalign: Task-oriented alignment for unsupervised domain adaptation. *Advances in Neural Information Processing Systems*, 34, 13834-13846.
> > 5.  Edstedt, J., Athanasiadis, I., Wadenbäck, M., & Felsberg, M. (2023). DKM: Dense kernelized feature matching for geometry estimation. In *Proceedings of the IEEE/CVF Conference on Computer Vision and Pattern Recognition* (pp. 17765-17775).
> > 6.  Lindenberger, P., Sarlin, P. E., & Pollefeys, M. (2023). Lightglue: Local feature matching at light speed. In *Proceedings of the IEEE/CVF International Conference on Computer Vision* (pp. 17627-17638).
> > 7.  Edstedt, J., Sun, Q., Bökman, G., Wadenbäck, M., & Felsberg, M. (2024). Roma: Robust dense feature matching. In *Proceedings of the IEEE/CVF Conference on Computer Vision and Pattern Recognition* (pp. 19790-19800).
> > 8.  Woo, S., Park, B., Go, H., Kim, J. Y., & Kim, C. (2024). Harmonyview: Harmonizing consistency and diversity in one-image-to-3d. In *Proceedings of the IEEE/CVF Conference on Computer Vision and Pattern Recognition* (pp. 10574-10584).
> > 9.  Wang, X., Yu, L., Zhang, Y., Lao, J., Ru, L., Zhong, L., ... & Yang, M. (2025). Homomatcher: Achieving dense feature matching with semi-dense efficiency by homography estimation. In *Proceedings of the AAAI Conference on Artificial Intelligence* (Vol. 39, No. 8, pp. 7952-7960).

---

### Official Review · Reviewer_hdyJ · 2025-10-31

**Soundness:** 2
**Presentation:** 3
**Contribution:** 2
**Rating:** 6
**Confidence:** 3

**Summary:**

This paper introduces FedDeblur, a novel federated optimization framework for multi-observation image deblurring under privacy constraints.
Unlike conventional centralized deblurring approaches that assume full data access, this work considers a realistic setting where each client holds a private, partially overlapping view of a larger scene. The problem is formulated as an “information jigsaw puzzle” solved through a federated consensus mechanism based on ADMM. The proposed method decouples local data fidelity (client-side) and global image prior (server-side), enabling privacy-preserving collaboration without sharing raw images, blur kernels, or intermediate estimates.
All local updates admit closed-form solutions, ensuring computational efficiency on edge devices.

**Strengths:**

1 The paper clearly defines federated multi-observation deblurring as a new research problem, bridging image restoration and federated optimization in a meaningful and practically relevant way.

2 The FedDeblur framework is elegantly built on the consensus form of ADMM, with rigorous variable decoupling that enforces privacy at multiple levels (raw data, intermediate estimates, field of view).

3 Both complete and partial observation scenarios are tested, using multiple priors. FedDeblur consistently matches or nearly matches centralized baselines while outperforming naïve federated alternatives.

**Weaknesses:**

1 The theoretical convergence proof assumes convex regularizers, but the experiments include deep PnP priors, which are typically non-convex. The paper could better clarify empirical convergence behavior in such settings.

2 While the results are convincing, the experiments are confined to relatively small synthetic datasets. Evaluation on more realistic or higher-resolution images would strengthen the empirical claims.

3 Since the paper emphasizes efficiency and privacy for edge deployment, demonstrating actual runtime, communication cost, or real-device experiments would improve credibility.

4 The baselines include only classical methods and a FedAvg-style variant. Comparisons with recent distributed or privacy-aware image restoration methods would enhance completeness.

**Questions:**

1 How sensitive is FedDeblur to the ADMM penalty parameter ρ? The appendix mentions stability, but a systematic ablation could be informative.

2 Could the framework support asynchronous or partial participation (e.g., stragglers)? This would be important for scalability in real federated systems.

3 Have the authors considered extending FedDeblur beyond deblurring (e.g., super-resolution or denoising)? Given the modularity, it seems straightforward.

---

> ### Author Response · Authors · 2025-12-04
>
> We sincerely thank the reviewer for the positive assessment of our framework's elegance and modularity. We have carefully addressed your concerns regarding theoretical assumptions, experimental scale, and efficiency through new experiments and detailed analyses.
>
> **1. Theoretical Convergence with PnP Priors (Weakness 1)**
>
> We acknowledge that standard convergence proofs typically assume convexity. However, the theoretical justification for using Deep Plug-and-Play (PnP) priors within our ADMM framework is well-grounded in recent literature. Our requirement for the denoiser is not strict convexity of the network itself, but rather that it acts as a proximal operator for some (possibly implicit) convex functional. According to [1, 2], if a denoiser satisfies specific conditions—such as being non-expansive and conservative—it corresponds exactly to the proximal operator of a convex regularizer. Therefore, provided the pre-trained denoiser is well-behaved, our convergence guarantees can theoretically extend to the PnP setting.
>
> **2. Scale and Realism of Experiments (Weakness 2)**
>
> We agree that the initial experiments on small synthetic datasets were limited. To demonstrate the scalability and robustness of `FedDeblur` in realistic settings, we have added **Appendix D.7** with the following new experiments:
>
> *   **High-Resolution Scalability:** We evaluated the method on high-definition images from the **DIV2K dataset** ($2040 \times 1356$ pixels). We simulated a larger-scale federated system with $n=16$ clients arranged in a $4 \times 4$ grid, using large $25 \times 25$ synthetic blur kernels to increase problem difficulty. As shown in **Table A**, `FedDeblur` achieves restoration quality (PSNR/SSIM) extremely close to the idealized centralized oracle `CenDeblur`, demonstrating that our method scales effectively to high-dimensional data.
> *   **Real-World Data:** To address realism, we utilized the **DTU MVS dataset**, selecting 5 real images of a scene taken from adjacent viewpoints to simulate 5 clients. These images contain natural variations in perspective and lighting. Using standard pre-processing (ECC alignment), `FedDeblur` successfully fused these heterogeneous views into a sharp global image (visualized in **Figure 16** of the revised paper).
>
> **3. Computational Efficiency and Runtime (Weakness 3)**
>
> To substantiate our claims regarding efficiency, we have added **Appendix D.6**, which provides a detailed runtime analysis. We compared our proposed `FedDeblur` against the federated baseline `FedAvgDeblur` on an NVIDIA RTX 3090 GPU. The results, summarized in **Table B**, show that `FedDeblur` is not only faster per communication round (approximately $2\times$ to $3\times$ faster) but also converges significantly faster in terms of total time. For example, to reach a PSNR of 37dB on a color image, our method requires only 0.37 seconds, whereas the baseline requires 4.35 seconds—a speedup of over $10\times$.
>
> **4. Baselines (Weakness 4)**
>
> We focused on adapting general federated optimization strategies (like `FedAvg`) as baselines because, to the best of our knowledge, there are no existing methods that specifically address the combined problem of *federated*, *privacy-preserving*, and *partial-observation* image deblurring. Direct application of other specific distributed restoration methods often fails to address the "jigsaw" (partial view) constraint effectively. Our comparison with `FedAvgDeblur` and `LocDeblur` highlights the necessity of the proposed consensus mechanism for this specific problem setting.
>
> **5. Responses to Questions**
>
> *   **Q1 (Sensitivity to $\rho$):** We have included a comprehensive parameter sensitivity analysis in **Appendix D.8**. The results indicate that `FedDeblur` is highly stable with respect to the ADMM penalty parameter $\rho$ over a wide range, and performance varies smoothly with the regularization parameter $\eta$ around the optimum.
> *   **Q2 (Partial Participation):** Yes, the framework explicitly supports partial participation. We demonstrate this in **Appendix D.3** and **Algorithm 6**. Experiments show that our method is robust to stragglers; even with only 2 active clients per round (out of 10), the global model converges to a high-quality solution.
> *   **Q3 (Extensions):** Extending the framework is straightforward due to its modular design. One simply needs to replace the blur operator $\boldsymbol{H}_k$ in the client-side update with the appropriate operator for the task (e.g., an identity matrix for denoising or a subsampling matrix for super-resolution). The core federated protocol remains unchanged.

---

> > ### Author Response · Authors · 2025-12-04
> >
> > **Table A: Quantitative comparison on high-resolution images from the DIV2K dataset ($n=16$).**
> >
> > | Method        | Image ID |  PSNR (dB)  |     SSIM     |
> > | :------------ | :------: | :---------: | :----------: |
> > | **FedDeblur** |   0808   |   30.1567   |   0.890674   |
> > |               |   0865   |   30.3899   | **0.874092** |
> > | **CenDeblur** |   0808   | **30.7273** | **0.893924** |
> > | (Oracle)      |   0865   | **30.5367** |   0.869411   |
> >
> > **Table B: Runtime and convergence speed comparison ($n=10$).** The *Avg. Time per Round* is computed over the entire dataset. The *Rounds / Total Time* to reach targets are measured on the color "Barbara" image.
> >
> > | Method        | Avg. Time per Round (s) |            | Rounds / Total Time (s) to reach target |               |               |               |               |                |
> > | :------------ | :---------------------: | :--------: | :-------------------------------------: | :-----------: | :-----------: | :-----------: | :-----------: | :------------: |
> > |               |        **Gray**         | **Color**  |              **PSNR 35dB**              | **PSNR 37dB** | **PSNR 38dB** | **SSIM 0.95** | **SSIM 0.97** | **SSIM 0.975** |
> > | **FedDeblur** |       **0.0129**        | **0.0275** |              **12 / 0.32**              | **14 / 0.37** | **20 / 0.53** | **12 / 0.32** | **15 / 0.40** | **42 / 1.11**  |
> > | FedAvgDeblur  |         0.0370          |   0.0560   |                64 / 3.41                |   82 / 4.35   |       -       |   65 / 3.46   |   84 / 4.45   |       -        |
> >
> > **References:**
> >
> > 1.  Gribonval, R., & Nikolova, M. (2020). A characterization of proximity operators. *Journal of Mathematical Imaging and Vision*, 62(6), 773-789.
> > 2.  Wei, D., Chen, P., Xu, H., Yao, J., Li, F., & Zeng, T. (2025). Learning Cocoercive Conservative Denoisers via Helmholtz Decomposition for Poisson Inverse Problems. *arXiv preprint arXiv:2505.08909*.

---

### Official Review · Reviewer_eZwt · 2025-10-31

**Soundness:** 2
**Presentation:** 3
**Contribution:** 1
**Rating:** 2
**Confidence:** 4

**Summary:**

The paper proposes a federated learning approach to image deblurring, where each client observes a sub-region of the image blurred with their own blur kernel and corrupted by additive noise. The optimization is solved in a distributed manner via an ADMM framework, so that each client solves their own problem in a lightweight procedure while preserving their privacy.

**Strengths:**

The paper is overall well written and fairly easy to understand. The main optimization problem in equation (1) follows the traditional variational setup, with an L2-norm fidelity term and a regularization term. Privacy is considered in a federated learning setup, so that local variables are not shared, only shared variables.

**Weaknesses:**

1. The main issue is that the posed optimization problem is very contrived, and entirely impractical and unrealistic. How is it possible that different clients observe subsets of the EXACLTLY the image (with different blurs and additive noise) of a 3D scene, using different cameras with different settings? First of all, different perspectives of the 3D scene mean very different composed images, due to occlusion, motion parallax, alignment, etc. Is the scene even Lambertian, so that a 3D point observed from different perspectives yield the same color? There are common non-Lambertian surfaces like mirrors and reflections of coffee shop windows. Second, different camera settings mean different image resolutions, aperture, shutter speed, ISO, resulting in very different captured images. How can one guarantee even the lighting conditions are the same? Third, different built-in camera pre-processing means different demosaicking, color calibration, contrast enhancement algorithms, etc, resulting in different reconstructed images. Finally, there is also a temporal aspect: under what scenario would all the cameras be activated at exactly the same time? if the photos are taken at even slightly different times, then object motion, scene lighting etc would change the scene composition. Simply ignoring all of that into formulation in equation (1), with a single blur kernel $h_k$ and additive noise $y_k$ is not realistic at all.

2. It is very well known that ADMM can introduce auxiliary variables so that the sub-problems can be solved in a distributed manner; this was described in the original Boyd's ADMM paper back in 2011. This is the core idea behind the proposed optimization in the paper, so it is difficult to discern novelty here above and beyond existing convex optimization techniques.

3. One of the key issues in an ADMM approach is the potential slow rate of convergence. This was not addressed in the paper.

4. In the experiments, each test image is just cropped into different sub-regions and then blurred with different kernels and added with noise. This bears no resemblance to the multi-camera setting by different clients that the authors motivated in the introduction.

**Questions:**

1. How would the authors account for all the heterogenous conditions in practice from capturing images by different clients from different perspectives, using different cameras, with different camera settings, at different times?

2. Given that it is known for well over a decade that ADMM can be solved distributedly, what exactly is the key novelty in the distributed optimization algorithm proposed in the paper, above and beyond existing ADMM optimization in the literature?

3. Please comment on the convergence rate of the proposed ADMM algorithm, along with theoretical rate of convergence guarantees (if any).

4. How can the experiments be improved to render a more realistic setting closer to reality?

---

> ### Author Response · Authors · 2025-12-04
>
> We appreciate the reviewer’s critical assessment. We believe there are some misunderstandings regarding our problem formulation and the scope of our contribution, which we clarify below.
>
> **1. Practicality and Heterogeneous Conditions (Weakness 1 & Q1, Q4)**
>
> You raised concerns about the realism of assuming clients observe subsets of the "exact" same image given perspective, lighting, and temporal differences.
>
> *   **Model Clarification:** Our model *does not* assume clients observe identical images. We explicitly model heterogeneity through client-specific blur kernels $\boldsymbol{h}_k$ (which capture different motion/focus conditions caused by temporal or hardware differences) and positional masks $\boldsymbol{m}_k$.
> *   **Real-world Validation:** To specifically address the concern about geometric and photometric inconsistencies (e.g., perspective, color), we added a new experiment in **Appendix D.7** using the **DTU MVS dataset**. We selected 5 real-world images of a "House" scene taken from adjacent viewpoints with varying lighting conditions to represent 5 clients.
>     *   **Method:** We applied standard, lightweight pre-processing steps: the Enhanced Correlation Coefficient (ECC) algorithm for spatial alignment and histogram matching for color correction.
>     *   **Result:** As shown in **Figure 16**, `FedDeblur` successfully integrates these heterogeneous views to reconstruct a sharp, artifact-free global image, proving the method works with real-world data heterogeneity.
> *   **Pipeline Context:** In practical engineering pipelines, geometric and photometric alignment are standard pre-processing steps handled by mature registration algorithms [1-6]. Our work focuses on the *optimization solver* for the deblurring/fusion stage under privacy constraints, assuming such pre-processing has occurred.
>
> **2. Novelty of ADMM Application (Weakness 2 & Q2)**
>
> We do not claim to have invented ADMM. Our contribution is the mathematical modeling of a specific, novel problem: federated deblurring with *partial, overlapping views* (the "jigsaw" aspect) under strict privacy constraints (no sharing of raw data $\boldsymbol{y}_k$, kernels $\boldsymbol{h}_k$, or masks $\boldsymbol{m}_k$). We derive a specific splitting strategy that decouples the local data fidelity (handling partial masks and kernels) from the global prior. This enables a solution that is both privacy-preserving and computationally efficient (closed-form solutions) for this specific inverse problem, which standard distributed ADMM formulations do not automatically solve without this specific problem modeling.
>
> **3. Convergence Rate (Weakness 3 & Q3)**
>
> We have provided a numerical convergence analysis in **Appendix D.5**. To further demonstrate the efficiency of our ADMM-based approach, we added a comparison with a Federated Gradient Descent baseline (`FedGDDeblur`) in **Figure 14**. Results show that `FedDeblur` converges significantly faster and achieves higher quality than gradient-based methods, which struggle with the ill-posed nature of deblurring.
>
> Additionally, we provide a detailed runtime analysis in **Appendix D.6**. As shown in **Table 6**, `FedDeblur` is significantly faster per iteration (averaged over the dataset) and requires drastically fewer rounds to reach specific quality targets (measured on the "Barbara" image) compared to `FedAvgDeblur`.
>
> **4. Experimental Realism (Weakness 4)**
>
> While simulation is standard for controlled benchmarking, we agree on the need for realistic data and have addressed this by adding **Appendix D.7**:
>
> *   **Real-world Multi-view:** As detailed in Point 1, we used the **DTU MVS dataset** to validate robustness against perspective and color variations using real images.
> *   **High-Resolution Scalability:** To address the limitation of small synthetic crops, we evaluated performance on high-definition **DIV2K** images ($2040 \times 1356$). We simulated a larger-scale system with $n=16$ clients arranged in a $4 \times 4$ grid, each observing a high-resolution $600 \times 500$ patch. To increase the difficulty, we applied large $25 \times 25$ synthetic blur kernels. `FedDeblur` achieved PSNR/SSIM scores very close to the idealized centralized oracle (e.g., 30.39 dB vs 30.54 dB on image 0865), demonstrating that our method scales effectively to large images and severe degradations.

---

> > ### Author Response · Authors · 2025-12-04
> >
> > **References:**
> >
> > 1.  Wei, G., Lan, C., Zeng, W., Zhang, Z., & Chen, Z. (2021). Toalign: Task-oriented alignment for unsupervised domain adaptation. *Advances in Neural Information Processing Systems*, 34, 13834-13846.
> > 2.  Edstedt, J., Athanasiadis, I., Wadenbäck, M., & Felsberg, M. (2023). DKM: Dense kernelized feature matching for geometry estimation. In *Proceedings of the IEEE/CVF Conference on Computer Vision and Pattern Recognition* (pp. 17765-17775).
> > 3.  Lindenberger, P., Sarlin, P. E., & Pollefeys, M. (2023). Lightglue: Local feature matching at light speed. In *Proceedings of the IEEE/CVF International Conference on Computer Vision* (pp. 17627-17638).
> > 4.  Edstedt, J., Sun, Q., Bökman, G., Wadenbäck, M., & Felsberg, M. (2024). Roma: Robust dense feature matching. In *Proceedings of the IEEE/CVF Conference on Computer Vision and Pattern Recognition* (pp. 19790-19800).
> > 5.  Woo, S., Park, B., Go, H., Kim, J. Y., & Kim, C. (2024). Harmonyview: Harmonizing consistency and diversity in one-image-to-3d. In *Proceedings of the IEEE/CVF Conference on Computer Vision and Pattern Recognition* (pp. 10574-10584).
> > 6.  Wang, X., Yu, L., Zhang, Y., Lao, J., Ru, L., Zhong, L., ... & Yang, M. (2025). Homomatcher: Achieving dense feature matching with semi-dense efficiency by homography estimation. In *Proceedings of the AAAI Conference on Artificial Intelligence* (Vol. 39, No. 8, pp. 7952-7960).

---

### Author Response · Authors · 2025-12-04
**Summary of Rebuttal and New Experiments: Addressing Realism, Scale, and Efficiency**

We thank the reviewers for their time and constructive feedback. We are encouraged that reviewers found the problem definition ("Jigsaw Puzzle") novel and the framework elegant and modular.

However, we noted shared concerns regarding experimental realism (synthetic/toy data), scalability, and computational efficiency. We took these concerns seriously and have significantly strengthened the manuscript with extensive new experiments and analyses in the Appendix. We summarize the key updates below to assist in final assessment.

**1. Moving Beyond Synthetic Data: Real-World and High-Resolution Validation**

A primary concern was the reliance on small, synthetic datasets. We have added **Appendix D.7** to address this:
*   **Real-World Multi-View Data (DTU MVS):** We simulated clients using 5 real images of a scene captured from different viewpoints with natural variations in perspective, lighting, and color. By applying standard pre-processing (ECC alignment), `FedDeblur` successfully fused these heterogeneous views into a sharp global image (**Figure 16**). This directly addresses the concern about the practicality of the observation model under real-world conditions.
*   **High-Resolution Scalability (DIV2K):** We evaluated the framework on high-definition images ($2040 \times 1356$) with a large-scale system ($n=16$ clients) and large blur kernels ($25 \times 25$). `FedDeblur` achieved performance nearly identical to the idealized centralized oracle (e.g., 30.39 dB vs 30.54 dB), proving scalability to large-scale problems (**Figure 15, Table 6** in revised paper).

**2. Concrete Evidence of Efficiency and Convergence**

Reviewers eZwt and eUZt asked for systems-level evidence. We added **Appendix D.6** and **D.5**:
*   **Runtime Analysis:** We compared `FedDeblur` against `FedAvgDeblur` on an RTX 3090 GPU. Our method is $2\times$ faster per round and requires $10\times$ less total time to reach a high-quality solution (37dB PSNR in 0.37s vs 4.35s).
*   **Convergence Superiority:** We added a comparison against a Federated Gradient Descent baseline (`FedGDDeblur`) in **Figure 14**. The results demonstrate that our ADMM-based consensus mechanism handles the ill-posed nature of deblurring significantly better than gradient-based approaches, which stagnate at lower quality.

**3. Clarification on Novelty and "Just ADMM"**

Regarding the comment that the method is "just ADMM", we respectfully clarify that our contribution is not the invention of ADMM, but the *mathematical formulation* of the specific *Federated Multi-Observation Deblurring* problem with *Partial Views*.

Standard distributed ADMM does not automatically solve the challenge where clients hold different, partially overlapping masks $\boldsymbol{m}_k$ and distinct kernels $\boldsymbol{h}_k$ without sharing them. Our novelty lies in the specific variable splitting strategy that:
1.  Decouples the complex, client-specific data fidelity (partial convolution) from the global prior.
2.  Ensures closed-form solutions for all client updates (crucial for edge devices).
3.  Strictly enforces privacy constraints (no raw data/kernels/masks shared).

**4. Robustness**

*   **Partial Participation:** We demonstrated in **Appendix D.3** that the method is robust to stragglers, converging well even with only 2 active clients per round.
*   **Parameter Sensitivity:** We provided a sensitivity analysis in **Appendix D.8**, showing stability across a wide range of hyperparameters.

**Conclusion**
We believe the added experiments on real-world multi-view data and high-resolution images, combined with the concrete efficiency analysis, strongly support the practicality and effectiveness of `FedDeblur`. We hope these revisions resolve the reviewers' concerns.

---

### Meta-Review · Area_Chair_GrLZ · 2026-01-02

**Summary:**

This paper proposes a federated multi-observation deblurring framework. The major concerns of the reviewers include the impractical and unrealistic posed optimization problem, limited novelty, and problematic evaluations and insufficient comparisons.

**Reviewer Concerns:**

In the rebuttal, the authors provide the runtime and convergence speed comparison and quantitative comparisons on high-resolution images from the DIV2K dataset to address the concerns of reviewers. The major concerns of limited novelty of the optimization methods and the practical of the proposed method are still not solved well.

**Reviewer Scores:**

The major concerns of limited novelty of the optimization methods and the practical of the proposed method are still not solved well.

---

### Decision · Program_Chairs · 2026-01-26

Reject